

# Predicted impacts of heterogeneous chemical pathways on particulate sulfur over Fairbanks, Alaska, the N. Hemisphere, and the Contiguous United States

Sara Farrell[1,2,3], Havala O. T. Pye[2], Robert Gilliam[2], George Pouliot[2], Deanna Huff[4], Golam Sarwar[2], William Vizuete[1], Nicole Briggs[5], Kathleen Fahey[2]

1. Department of Environmental Science and Engineering, The University of North Carolina at Chapel Hill, Chapel Hill, NC 27516, USA

2. Office of Research and Development, U.S. Environmental Protection Agency, Research Triangle Park, Durham, NC 27709, USA

3. Oak Ridge Institute for Science and Education, U.S. Environmental Protection Agency, Research Triangle Park, NC 27709, USA

4. Alaska Department of Environmental Conservation, P.O. Box 111800, Juneau, 99811-1800

5. Laboratory Services and Applied Sciences Division at USEPA, Region 10, Seattle, WA 98101

*Correspondence to*: Kathleen Fahey (Fahey.Kathleen@epa.gov) and Sara Farrell (Farrell.Sara@epa.gov)

**Abstract.** A portion of Alaska's Fairbanks North Star Borough was designated as nonattainment for the 2006 24-hour $PM_{2.5}$ National Ambient Air Quality Standard (NAAQS) in 2009. $PM_{2.5}$ NAAQS exceedances in Fairbanks mainly occur during

the dark and cold winters, when temperature inversions form and trap high emissions at the surface. Sulfate ($SO_4^{2-}$), often the second largest contributor to $PM_{2.5}$ mass during these wintertime PM episodes, is underpredicted by atmospheric chemical transport models (CTMs). Most CTMs account for primary $SO_4^{2-}$, and secondary $SO_4^{2-}$ formed via gas-phase oxidation of sulfur dioxide ($SO_2$) and in-cloud aqueous oxidation of dissolved S(IV). Heterogeneous sulfur chemistry in aqueous aerosols, not often included in CTMs, may help better represent the high $SO_4^{2-}$ concentrations observed during Fairbanks

winters. In addition, hydroxymethanesulfonate (HMS), a particulate sulfur species sometimes misidentified as $SO_4^{2-}$, is known to form during Fairbanks winters. Heterogeneous formation of $SO_4^{2-}$ and HMS in aerosol liquid water (ALW) was implemented in the Community Multiscale Air Quality (CMAQ) modeling system. CMAQ simulations were performed for wintertime PM episodes in Fairbanks (2008) as well as over the N. Hemisphere and Contiguous United States (CONUS) for 2015-2016. The added heterogeneous sulfur chemistry reduced model mean sulfate bias by ~0.6 µg/m³ during a cold winter

PM episode in Fairbanks, AK. Improvements in model performance are also seen in Beijing, during wintertime haze events (reducing model mean sulfate bias by ~2.7 µgS/m³). This additional sulfur chemistry also improves modeled summertime $SO_4^{2-}$ bias in the southeast U.S. with implications of future modeling of biogenic organosulfates.

## 1 Introduction

Radiative forcing and climate effects attributed to $PM_{2.5}$ (fine particulate matter 2.5 µm or less in diameter) remain among

the most uncertain in climate change assessments (IPCC, 2013). Acute and long-term exposure to $PM_{2.5}$ has been associated with negative health outcomes including, but not limited to, acute myocardial infarction, stroke, and respiratory



complications (Chen et al., 2018; Hayes et al., 2020; Silva et al., 2016; USEPA, 2019; Yang et al., 2021; Yitshak-Sade et al., 2018). Some of the most extreme $PM_{2.5}$ episodes in history have occurred when inverted vertical temperature profiles cause stable atmospheric conditions that limit pollution dilution (Holzworth, 1972; Malek et al., 2006; Scott, 1953; Wallace and

Kanaroglou, 2009). Surface based temperature inversions are characteristic of Fairbanks and North Pole, Alaska winters, and are associated with degraded air quality in this region (Malingowski et al., 2014; Mayfield and Fochesatto, 2013). Wintertime $PM_{2.5}$ episodes have impacted human health in these cities (McLaughlin, 2010), with this region exceeding the National Ambient Air Quality Standards (NAAQS) for $PM_{2.5}$ since 2009, when portions of the Fairbanks North Star Borough were designated as nonattainment for the 2006 24-hour $PM_{2.5}$ NAAQS (ADEC, 2017, 2019).

Sulfate ($SO_4^{2-}$), often a major component of $PM_{2.5}$ in Fairbanks and North Pole (ADEC, 2017) as well as globally (Snider et al., 2016), can be emitted directly (primary) or formed via gas-phase oxidation of sulfur dioxide ($SO_2$) (Calvert et al., 1978), particle surface oxidation of $SO_2$ (Clements et al., 2013; Wang et al., 2021), and aqueous-phase oxidation of inorganic sulfur species with oxidation number 4 (S(IV) = $SO_2 \cdot H_2O$ + $HSO_3$ + $SO_3$) (secondary) (Hoffmann and Calvert, 1985; Ibusuki and

Takeuchi, 1987; Lagrange et al., 1994; Lee and Schwartz, 1983a; Maahs, 1983; Maaß et al., 1999; Martin and Good, 1991; McArdle and Hoffmann, 1983). Aside from contributing directly to $PM_{2.5}$ mass, $SO_4^{2-}$ can facilitate the formation of other $PM_{2.5}$ species as a reactant (Brüggemann et al., 2020; Huang et al., 2019; Huang et al., 2020; Surratt et al., 2010), by increasing aerosol water uptake (Kim et al., 1994; Nguyen et al., 2014), and by altering aerosol acidity (Li et al., 2022; Pye et al., 2020).


Under heavily polluted haze conditions, such as those common in the North China Plain during the winter, recent studies have suggested that secondary $SO_4^{2-}$ may be efficiently produced in aerosol liquid water (ALW) (Cheng et al., 2016; Fan et al., 2020; Liu et al., 2020). Hygroscopic $PM_{2.5}$ (both inorganic and organic) can increase ALW content (Nguyen et al., 2014; Petters and Kreidenweis, 2007; Pye et al., 2017), which can facilitate secondary $SO_4^{2-}$ formation (Zhang et al., 2021a),

enhancing $SO_4^{2-}$ concentrations in a positive feedback loop – which is particularly important during high relative humidity haze events (Cheng et al., 2016; Song et al., 2021b; Wang et al., 2016; Wang et al., 2014).

Chemical transport models generally include secondary $SO_4^{2-}$ formation via gas-phase oxidation of $SO_2$ by OH, and in-cloud aqueous-phase oxidation of dissolved S(IV) species by oxidants such as hydrogen peroxide ($H_2O_2$), ozone ($O_3$), peroxyacetic

acid (PAA), methyl hydroperoxide (MHP), and/or oxygen ($O_2$) catalyzed by transition metal ions (TMI-$O_2$), iron ($Fe^{3+}$) and manganese ($Mn^{2+}$). Limited to these formation pathways, CTMs have been unable to reproduce the high levels of $SO_4^{2-}$ observed during wintertime PM events in Beijing and the sub-Arctic. This persistent underprediction suggests that CTMs are lacking $SO_4^{2-}$ formation pathways that take place when photochemistry and cloud liquid water are limited (Eckhardt et al., 2015; Gao et al., 2016; Wang et al., 2014). Previous studies have suggested that heterogeneous sulfate formation in ALW

may account for at least part of this underprediction (Wang et al., 2014; Zheng et al., 2015). Wang et al. (2014) implemented



generalized heterogeneous reactive uptake of $SO_2$ to form $SO_4^{2-}$ in ALW (where relative humidity-dependent uptake coefficients were specified rather than calculated) in the GEOS-Chem global model (Chen et al., 2009), and this led to improved model-observation comparisons for $SO_4^{2-}$ during a wintertime haze event over North China (Wang et al., 2014). Similarly, when Zheng et al. (2015) implemented generalized heterogeneous sulfur chemistry in WRF-CMAQ, the $SO_4^{2-}$

normalized mean bias decreased from -54.2% to 6.3% for Beijing haze events (Zheng et al., 2015).

The inclusion of hydroxymethanesulfonate (HMS) chemistry in CTMs may also ameliorate negative model bias. General $SO_4^{2-}$ PM measurement methods struggle to disentangle $SO_4^{2-}$ spectra from that of inorganic S(IV) and HMS (Dovrou et al., 2019). HMS is formed from the aqueous-phase reaction of S(IV) with formaldehyde (HCHO) (Boyce and Hoffmann, 1984;

Deister et al., 1986; Kok et al., 1986; Kovacs et al., 2005; Olson and Hoffmann, 1986). Recent field, modeling, and experimental studies have highlighted the importance of secondary $SO_4^{2-}$ and HMS formation in aqueous aerosols during wintertime haze events (Campbell et al., 2022; Dovrou et al., 2019; Ma et al., 2020; Moch et al., 2020; Song et al., 2019). Moch et al. (2020) suggested up to 25% of measured $SO_4^{2-}$ may actually be HMS in heavily polluted regions. Arctic and sub-arctic winter conditions may favor HMS formation, as colder temperatures increase the solubility of $SO_2$ and HCHO, and

limited sunlight reduces the photo-oxidation of HCHO (Moch et al., 2020; Pandis and Seinfeld, 1989; Sander, 2015; Song et al., 2019; Staudinger and Roberts, 1996). In the work of Song et al. (2021) including generalized heterogeneous cloud HMS chemistry in GEOS-Chem reduced model-measurement differences in HMS/$SO_4^{2-}$ ratios. The inclusion of heterogeneous formation and loss of HMS in ALW increased modelled HMS concentrations, particularly in China (Song et al., 2021a).

Many laboratory studies characterizing rate coefficients and expressions for aqueous phase $SO_4^{2-}$ formation were performed under dilute conditions, characteristic of cloud droplets. The ionic strength of aerosol particles, however, can be several orders of magnitude higher than that of cloud droplets due to aerosols containing much lower concentrations of water (Mekic and Gligorovski, 2021). Ionic strength has been found to impact the aqueous-phase formation of $SO_4^{2-}$, increasing the rate of aqueous-phase kinetics for $NO_2$ and $H_2O_2$ oxidation of S(IV) (Chen et al., 2019; Liu et al., 2020) and inhibiting the aqueous-

phase kinetics of TMI-catalyzed $O_2$ oxidation (Ibusuki and Takeuchi, 1987; Martin and Good, 1991; Martin and Hill, 1987). Experimental studies have also shown that high ionic strength may increase or decrease (effective) Henry's law coefficients of reactants compared to pure water (Ali et al., 2014; Chen et al., 2019; Kosak-Channing and Helz, 1983; Lagrange et al., 1994; Liu et al., 2020; Millero et al., 1989; Shao et al., 2019).

In this paper we describe the implementation of heterogenous sulfur chemistry in ALW in the Community Multiscale Air Quality (CMAQv5.3.2) modeling system (USEPA, 2020), leading to additional $SO_4^{2-}$ and HMS formation. In addition to heterogeneous chemistry updates, ionic strength effects were added to condensed-phase rate expressions and Henry's law coefficients of some species. The updated model was applied for several time periods and for different domains and horizonal resolutions. Two historical wintertime PM episodes were simulated for a finely resolved (1.33 km) domain





centered over Fairbanks, Alaska, winter, and summer periods over CONUS (12 km) during 2016, and the 2015-2016 winter

season over the N. hemisphere (108 km) to investigate the impacts of these updates for different chemical regimes, domains,

and seasons. Changes to $SO_4^{2-}$, HMS, and $SO_4^{2-}$ + HMS (PM$_{2.5,\text{sulf}}$) predictions were tracked with each update (i.e., for (1)

adding heterogeneous sulfur reactions and (2) adding ionic strength effects), and model performance was evaluated with

available observations. This study aims to better understand the impacts that heterogeneous sulfur chemistry

parameterizations may have on predicted PM$_{2.5,\text{sulf}}$ concentrations and whether the additional chemistry can resolve $SO_4^{2-}$

underpredictions in cold and dark conditions.

## 2 Methods

### 2.1 Heterogeneous Sulfur Chemistry

In this study, reactions that transform $SO_2$ to $SO_4^{2-}$ and HMS are simulated in both cloud and ALW. In ALW, the production

of particulate $SO_4^{2-}$ and HMS is parameterized as a set of first-order heterogeneous reactions of $SO_2$ (gas) or reactants (e.g.,

HCHO, O₃, NO₂, etc.) as follows:

$$P_{SO2,reactant} \xrightarrow{k_{het}} PM_{2.5,sulf} \qquad \text{(Reaction. 1)}$$

Where PM$_{2.5,\text{sulf}}$ refers to both sulfate and HMS and $k_{het}$ is the heterogeneous rate constant (Eq. 1), which accounts for gas-

to-particle mass transfer processes, aqueous reactive uptake and sulfur transformations (Jacob, 2000).

$$k_{het}(s^{-1}) = \frac{SA}{\frac{r_p}{D_g} + \frac{4}{v\gamma}} \qquad \text{(Eq. 1)}$$

where $SA$ is the aerosol surface area (m²/m³), $r_p$ is the effective particle radius (m), $D_g$ is gas-phase diffusivity of a reactant

(m²/s), and $v$ is the mean molecular speed of the partitioning gas (m/s). $\gamma$ is the reactive uptake coefficient and is given by

the following equation (Hanson et al., 1994; Jacob, 2000; Schwartz, 1986):


$$\gamma = \left[\frac{1}{\alpha} + \frac{v}{4HRT\sqrt{D_a k_{chem}}} \cdot \frac{1}{\coth(q) - \frac{1}{q}}\right]^{-1} \qquad \text{(Eq. 2)}$$

where $\alpha$ is the mass accommodation coefficient of a species, $H$ is the effective Henry's law coefficient (M/atm) at

temperature $T$ (K), $R$ is the ideal gas constant, 0.08206 (L·atm/mol·K), $D_a$ is the aqueous-phase diffusivity (assumed here as





$10^{-9}$ m²/s), $k_{chem}$ represents the pseudo-first order condensed-phase rate coefficient (s⁻¹) (Table 1), and q is the diffuso-reactive parameter defined as $q = r_p \sqrt{\frac{k_{chem}}{D_a}}$ (Schwartz and Freiberg, 1981). The rate expressions in aerosol water for the

base configuration (hereafter referred to as "Base_Het") are the same as those used for cloud chemistry (Appel et al., 2017; Sarwar et al., 2013). Heterogeneous sulfur chemistry was calculated when relative humidity (RH) was greater than or equal 50% following Shao et al. (2019), assuming that below a 50% RH, the aerosol water content would be too low for heterogeneous sulfur chemistry to take place (Sun et al., 2013). The heterogeneous chemistry is also solved simultaneously with gas-phase chemistry.

**2.2 Accounting for ionic strength effects and alternative chemical rate expressions**

Several studies have investigated the impact of high ionic strength on sulfate oxidation rates originally developed for dilute conditions. Depending on reaction pathway, rates may be enhanced or diminished with increased ionic strength (Chen et al., 2019; Lagrange et al., 1994; Liu et al., 2020; Shao et al., 2019). Ionic strength is calculated in CMAQ as:

$$I = \frac{1}{2} \sum_{i=1}^{n} m_i z_i^2 \qquad \text{(Eq. 3)}$$

where $m_i$ is the solute concentration (M) in aerosol or cloud liquid water and $z_i$ is the charge associated with each modelled ion.

SO₂ can be oxidized in the aqueous phase by O₂ when catalyzed by TMI (specifically $Fe^{3+}$ and $Mn^{2+}$) with synergies existing when both $Fe^{3+}$ and $Mn^{2+}$ are present (Altwicker and Nass, 1983; Ibusuki and Takeuchi, 1987; Martin and Good, 1991). This

reaction pathway has been found to be an important secondary $SO_4^{2-}$ formation pathway, especially at low pH and when photochemistry is limited and Fe is more soluble (Cheng et al., 2016; Li et al., 2020; Liu et al., 2022; Song et al., 2021b). Ibusuki and Takeuchi (1987) found that the rate of S(IV) oxidation by the TMI-O₂ pathway peaked around pH = 4.2, decreased with decreased temperature, and was enhanced by high concentrations of TMI (Ibusuki and Takeuchi, 1987). Martin and Good investigated the impact of higher S(IV) concentrations on this $SO_4^{2-}$ formation pathway and found that

higher S(IV) concentrations alter the rates of catalysis via $Fe^{3+}$ and $Mn^{2+}$ along with their synergistic catalysis and did not explore a pH nor temperature dependency (Martin and Good, 1991). However, at similar pH ranges, S(IV) and soluble $Fe^{3+}$ and $Mn^{2+}$ concentrations, Martin and Good found that their rate expression agreed well with that by Ibusuki and Takeuchi (Ibusuki and Takeuchi, 1987; Martin and Good, 1991). When Shao et al. (2019) implemented the Ibusuki-Takeuchi TMI-catalyzed O₂ oxidation rate expression in ALW in GEOS-Chem, they found that this pathway accounted for 67-69% of $SO_4^{2-}$

formed over China (Shao et al., 2019). The presence of higher S(IV) concentrations, however, may warrant the use of a $SO_4^{2-}$ formation rate that takes into consideration faster rates of TMI catalysis (Martin and Good, 1991). The TMI-O₂ oxidation





pathway from Martin and Good (1991) is used in the "Base_Het" simulation and the TMI-$O_2$ oxidation pathway from Ibusuki and Takeuchi (1987) is used in the "TMI_sens" simulation to explore the range in $SO_4^{2-}$ formation possible by this pathway (Table 1). For both implementations of this pathway, ionic strength impacts were added as high ionic strength has

been found to limit the TMI-catalyzed oxidation of S(IV) to $SO_4^{2-}$ (Martin and Hill, 1967; Martin and Hill, 1987).

While the TMI-catalyzed oxidation pathway can be a significant contributor to secondary $SO_4^{2-}$ formation, especially at low pH, some studies suggest $SO_4^{2-}$ formation during Beijing winter haze events may be dominated by the reaction of $SO_2$ and $NO_2$ in aerosol and/or cloud and fog water under mildly acidic or neutral conditions (Cheng et al., 2016; Wang et al., 2020; Yang et al., 2019). This oxidation pathway may increase in importance with increasing ionic strength (Cheng et al., 2016).

Recent chamber work by Chen et al. (2019) found that increasing ionic strength increased the rate of secondary $SO_4^{2-}$ formation from $NO_2$ oxidation (Chen et al., 2019). To assess the potential impact ionic strength may have on this pathway, the ionic strength dependent $NO_2$ oxidation rate of Chen et al., (2019) was included in the "TMI_NO2" sensitivity simulation (Table 1).

While the aforementioned pathways are pH dependent, $SO_2$ aqueous oxidation by $H_2O_2$ is pH independent for pH > 2 due to the opposing dependencies of the reaction rate coefficient and S(IV) solubility on pH (Clifton et al., 1988; Seinfeld and Pandis, 2016). This pathway has been studied extensively in dilute conditions representative of cloud droplets (Hoffmann and Calvert, 1985; Maaß et al., 1999; McArdle and Hoffmann, 1983). Maaß et al. (1999) found that oxidation of S(IV) by $H_2O_2$ increases with increased ionic strength and formulated a semi-empirical relationship between ionic strength and the

reaction rate coefficient for the S(IV)-$H_2O_2$ oxidation pathway (upper limit of ionic strength in this study = 5M) (Maaß et al., 1999). Field measurements during Chinese haze events have reported ionic strengths of aerosols ranging from 14-43 M (Cheng et al., 2016; Fountoukis and Nenes, 2007). Liu et al. (2020) recently studied this pathway using aerosol flow reactors to determine an ionic strength-based enhancement factor that encapsulates the combined ionic strength effects on Henry's law coefficients, dissociation, and condensed-phase kinetics. This study found that increasing ionic strength from 0 to 14 M

resulted in an order-of-magnitude increase in $SO_4^{2-}$ production rate (Liu et al., 2020).

$O_3$ is another important aqueous-phase oxidant of $SO_2$ with a reaction rate that increases with increasing pH (Maahs, 1983) and ionic strength (Lagrange et al., 1994). The ionic strength enhancement factor for this rate has been implemented with other ionic strength enhancement or inhibition factors in the "All_Ionic" sensitivity simulation (Table 1).


**Table 1. Pseudo-first-order rate constants ($k_{chem}$) and ionic strength ($I$) factors for each simulation.**

| Uptake gas | Model Simulation | $k_{chem}$ ($s^{-1}$) | Product | Reference |
|---|---|---|---|---|
| $O_3$ | Base_Het, | $k_{O3} = k_1[H_2SO_3] + k_2[HSO_3^-] + k_3[SO_3^{2-}]$ | $SO_4^{2-}$ | (Hoffmann and |



| | TMI_sens, TMI_NO2_sens | $k_1 = 2.4 \times 10^4 \ \mathrm{M^{-1}s^{-1}}$ <br><br> $k_2 = 3.7 \times 10^5 \times e^{-5530 \times \left(\frac{1}{T} - \frac{1}{298}\right)} \ \mathrm{M^{-1}s^{-1}}$ <br><br> $k_3 = 1.5 \times 10^9 \times e^{-5280 \times \left(\frac{1}{T} - \frac{1}{298}\right)} \ \mathrm{M^{-1}s^{-1}}$ | | Calvert, 1985) |
|---|---|---|---|---|
| | All_Ionic | $k_{O3,I} = k_{O3} \times (1 + (b \times I_{O_3}))$ <br><br> $I_{O_3,max} = 1.2 \ \mathrm{M}$ <br><br> $b = 1.94$[a] | $SO_4^{2-}$ | (Lagrange et al., 1994) |
| $H_2O_2$ | Base_Het, TMI_sens, TMI_NO2_sens | $k_{H2O2} = \frac{k_4[\mathrm{H^+}][\mathrm{HSO_3^-}]}{1 + K[\mathrm{H^+}]}$ <br><br> $k_4 = 7.45 \times 10^7 \times e^{-4430 \times \left(\frac{1}{T} - \frac{1}{298}\right)} \ \mathrm{M^{-1}s^{-1}}$ <br><br> $K = 13\mathrm{M^{-1}}$ | $SO_4^{2-}$ | (McArdle and Hoffmann, 1983) |
| | All_Ionic | $k_{H2O2,I} = k_4 \times [HSO_3^-] \times [H^+] \times$ <br><br> $10^{0.36 I_{H_2O_2} - \frac{1.018\sqrt{I_{H_2O_2}}}{1 + 0.17\sqrt{I_{H_2O_2}}}}$ <br><br> $k_4 = 9.1 \times 10^7 \times e^{-3572 \times \left(\frac{1}{T} - \frac{1}{298}\right)} \ \mathrm{M^{-1}s^{-1}}$ <br><br> $I_{H_2O_2,max} = 5\mathrm{M}$ | $SO_4^{2-}$ | (Maaß et al., 1999) |
| $NO_2$ | Base_Het, TMI_sens | $k_{NO2} = k_5[\mathrm{S(IV)}]$ [b] <br> $k_5 = 2 \times 10^6 \ \mathrm{M^{-1}s^{-1}}$ | $SO_4^{2-}$ | (Lee and Schwartz, 1983b) |
| | TMI_NO2_sens, All_Ionic | $k_{NO2,I} = k_{5,ionic}[\mathrm{S(IV)}]$ [b] <br><br><br> $k_{5,ionic} = 10^{6.1 + \frac{3.1 \times I_{NO_2}}{I_{NO_2} + 0.2}}$ <br><br> $I_{NO_2,max} = 1.14\mathrm{M}$ | $SO_4^{2-}$ | (Chen et al., 2019) |
| $SO_2$ | Base_Het | $k_{TMIO2,I} =$ <br><br> $\frac{k_6[\mathrm{Mn(II)}] + k_7[\mathrm{Fe(III)}] + k_8[\mathrm{Mn(II)}][\mathrm{Fe(III)}]}{1 + 75 \times S(VI)_S^{0.67}} \times$ <br><br> $10^{b \times \frac{\sqrt{I_{TMI}}}{1 + \sqrt{I_{TMI}}}}$ <br><br> $k_6 = 750 \ \mathrm{M^{-1}s^{-1}}$ <br><br> $k_7 = 2600 \ \mathrm{M^{-1}s^{-1}}$ | $SO_4^{2-}$ | (Martin and Good, 1991) |





| | | | | |
|---|---|---|---|---|
| | | $k_8 = 1e^{10} \text{M}^{-2}\text{s}^{-1}$ <br><br> $I_{TMI,max} = 2\text{M}$ <br><br> $b = -4.07$ [c] | | (Martin and Hill, 1987) |
| | TMI_sens, <br> TMI_NO2_sens, <br> All_Ionic | $k_{TMIO2,I} = k_6[\text{H}^+]^{-0.74}[\text{Mn(II)}][\text{Fe(III)}]$ <br><br> $(for\ \text{pH} \leq 4.2)$ <br><br> $k_6 = 3.72 \times 10^7 \times e^{-8431.6 \times \left(\frac{1}{T} - \frac{1}{297}\right)} \times$ <br><br> $10^{b \times \frac{\sqrt{I_{TMI}}}{1+\sqrt{I_{TMI}}}} \text{M}^{-2}\text{s}^{-1}$ <br><br> $k_{TMIO2,I} = k_7[\text{H}^+]^{0.67}[\text{Mn(II)}][\text{Fe(III)}]$ <br><br> $(for\ \text{pH} > 4.2)$ <br><br> $k_7 = 2.51 \times 10^{13} \times e^{-8431.6 \times \left(\frac{1}{T} - \frac{1}{297}\right)} \times$ <br><br> $10^{b \times \frac{\sqrt{I_{TMI}}}{1+\sqrt{I_{TMI}}}} \text{M}^{-2}\text{s}^{-1}$ <br><br> $I_{TMI,max} = 2\text{M}$ <br><br> $b = -4.07$ [c] | $SO_4^{2-}$ | (Ibusuki and Takeuchi, 1987) <br><br><br><br><br><br><br><br><br> (Martin and Hill, 1987) |
| HCHO[d] | Base_Het, <br> TMI_sens, <br> TMI_NO2_sens <br> All_Ionic | $k_8[\text{HSO}_3^-] + k_9[\text{SO}_3^{2-}]$ <br><br> $k_8 = 790 \times e^{-2900 \times \left(\frac{1}{T} - \frac{1}{298}\right)}$ <br><br> $k_9 = 2.5 \times 10^7 \times e^{-2450 \times \left(\frac{1}{T} - \frac{1}{298}\right)}$ | HMS | (Boyce and Hoffmann, 1984) |

[a] The ionic strength cap chosen corresponds with the activation energy for rate constant, $k_1$, which matches the ionic strength cap, $b$, for sodium perchlorate (NaClO$_4$).

[b] $[S(IV)](M) = [SO_2 \cdot H_2O] + [HSO_3^-] + [SO_3^{2-}]$

[c] $b$ value corresponds to the factor associated with Mn-catalyzed O$_2$ oxidation of SO$_2$, corresponding to $\geq 10^{-4}$ M sulfur (Martin and Hill, 1987).

[d] Also included is the loss of HMS from decomposition and OH oxidation (Song et al., 2021)






**Table 2. Henry's law coefficients used per model simulation.**

| Species | Model Simulation | $H_A$ (M atm$^{-1}$) | Reference |
|---|---|---|---|
| O$_3$ | Base_Het TMI_sens TMI_NO2_sens | $H_{O_3}^{I_{O_3}=0} = 0.0114 \times e^{2300\times(\frac{1}{T}-\frac{1}{298})}$ | (Kosak-Channing and Helz, 1983) |
| | All_Ionic | $H_{O_3} = e^{\frac{2297}{T}-2.659\times I_{O_3}+688\times\frac{I_{O_3}}{T}-12.19}$ $I_{O_3,max} = 0.6\ M$ | (Kosak-Channing and Helz, 1983) |
| H$_2$O$_2$ | Base_Het TMI_sens TMI_NO2_sens | $H_{H_2O_2}^{I_{H_2O_2}=0} = 1.1 \times 10^5 \times e^{7400\times(\frac{1}{T}-\frac{1}{298})}$ | (O'Sullivan et al., 1996) |
| | All_Ionic | $H_{H_2O_2}^{I_{H_2O_2}=0} = 1.3 \times 10^5 \times e^{7300\times(\frac{1}{T}-\frac{1}{298})}$ $\frac{H_{H_2O_2}}{H_{H_2O_2}^{I_{H_2O_2}=0}} = 1 - 1.414 \times 10^{-3} \times I_{H_2O_2}^2 + 0.121 I_{H_2O_2}$ $I_{H_2O_2,max} = 5M$ | (Seinfeld and Pandis, 2016) (Ali et al., 2014; Liu et al., 2020) |
| NO$_2$ | Base_Het, TMI_sens TMI_NO2_sens All_Ionic | $H_{NO_2} = 0.012 \times e^{2500\times(\frac{1}{T}-\frac{1}{298})}$ | (Chameides, 1984) |
| SO$_2$ | Base_Het TMI_sens TMI_NO2_sens | $[H_2SO_3] = H_{SO_2} \times p_{SO_2}$ $[HSO_3^-] = K_{a1} \times [H_2SO_3]/[H^+]$ $[SO_3^{2-}] = K_{a2} \times [HSO_3^-]/[H^+]$ $H_{SO_2} = 1.4 \times e^{2900\times(\frac{1}{T}-\frac{1}{298})}$ $K_{a1} = 0.013 \times e^{1960\times(\frac{1}{T}-\frac{1}{298})}$ $K_{a2} = 6.6 \times 10^{-8} \times e^{1500\times(\frac{1}{T}-\frac{1}{298})}$ | (Lide et al., 1995) |
| | All_Ionic [a] | $H_{SO_2}^{I_{SO_2}=0} = 1.2 \times e^{3100\times(\frac{1}{T}-\frac{1}{298})}$ | (Seinfeld and Pandis, 2016) |





| | | | |
|---|---|---|---|
| | | $\frac{H_{SO_2}}{H_{SO_2}^{I_{SO_2}=0}} = 10^{\left(\left(\frac{22.3}{T}-0.0997\right)\times I_{SO_2}\right)}$ $K_{a1}^I = K_{a1} \times 10^{0.5\sqrt{I}-0.31I}$ $K_{a2}^I = K_{a2} \times 10^{1.052\sqrt{I}-0.36I}$ $I_{SO_2,max} = 6\,\text{M}$ | (Millero et al., 1989) |
| HCHO | Base_Het, TMI_sens TMI_NO2_sens All_Ionic | $H_{HCHO} = \frac{H_{HCHO}^*}{1+K_{HCHO}^{HYD}}$ $H_{HCHO}^* = 3200 \times e^{6800\times\left(\frac{1}{T}-\frac{1}{298}\right)}$ $K_{HCHO}^{HYD} = \frac{0.18\times e^{4030\times\left(\frac{1}{T}-\frac{1}{298}\right)}\times 55.5M}{0.0051}$ | (Seinfeld and Pandis, 2016) (Ervens et al., 2003; Staudinger and Roberts, 1996) |

[a] Aqueous phase concentrations of S(IV) are calculated similarly to the Base_Het, TMI_sens, and TMI_NO2_sens but with ionic-strength dependent equilibrium coefficients.

**2.3 Model base case and sensitivity simulations**

Several CMAQ configurations were used here to understand the impacts of adding heterogeneous sulfur chemistry, ionic strength, and the use of alternative pseudo-first order rate expressions. A base case CMAQ simulation ("Base") was completed using in-cloud $SO_4^{2-}$ formation from aqueous oxidation by $H_2O_2$, $O_3$, PAA, MHP, and via TMI-catalyzed $O_2$ of $SO_2$ and gas-phase oxidation of $SO_2$ by OH (Fahey et al., 2017; Sarwar et al., 2013).

To account for the impacts of heterogeneous sulfur chemistry in ALW, the Base_Het model simulation was completed for all domains (see Model Configuration) using the aforementioned heterogenous reactive uptake parameterizations (Eq. 1-2). Parameters from CMAQ's KMT2 cloud chemistry model (Fahey et al., 2017; Fahey et al.) were used to calculate $k_{chem}$ (Table 1) and g. An ionic strength inhibition term was added to the TMI-catalyzed $O_2$ pathway in aerosol water to account for the limiting effect of ionic strength on this pathway. The ionic strength was capped at $I_{max} = 2\,\text{M}$ to reflect experimental
constraints (Martin and Good, 1991; Martin and Hill, 1987; Seinfeld and Pandis, 2016). For this pathway in both cloud and ALW, $Fe^{3+}$ was assumed to be 90% of dissolved Fe at night and 10% during the daytime, with soluble fractions of Mn and Fe assumed to be 0.5 and 0.1 of total Fe and Mn respectively (Alexander et al., 2009). Other sources of less soluble Fe and Mn emissions, such as dust, are likely minimal given snow-cover for this domain and episode (Shao et al., 2019).



The $k_{chem}$ for the TMI-catalyzed $O_2$ pathway in the Base_Het case is neither temperature nor pH dependent (Martin and Good, 1991; Martin and Hill, 1987). Ibusuki and Takeuchi (1987) found both a pH and temperature dependence on the $k_{chem}$ for this pathway (Ibusuki and Takeuchi, 1987; Martin and Hill, 1987). To explore the effects of both a pH and temperature dependences on the rate of TMI-catalyzed sulfur oxidation pathway, a sensitivity simulation, "TMI_sens", was run (Ibusuki and Takeuchi, 1987). Given that this $k_{chem}$ is reduced in colder temperatures, the TMI_sens run likely

represents a lower bound on $SO_4^{2-}$ formation for this pathway during winter episodes. This $k_{chem}$ also uses the same solubility, dissociation and ionic strength bounds as the TMI-catalyzed $O_2$ oxidation pathway used in the Base_Het simulation.

        In the "TMI_NO2_sens" simulations both the alternative $k_{chem}$ for the TMI-catalyzed $O_2$ pathway and an ionic strength-

dependent $k_{chem}$ for the $NO_2$ oxidation pathway in ALW are included (Chen et al., 2019; Ibusuki and Takeuchi, 1987). Both the TMI-$O_2$ and the $NO_2$ oxidation $k_{chem}$'s favor weakly acidic pH regimes (Cheng et al., 2016; Martin and Good, 1991). This sensitivity simulation was implemented to analyze potential competition between two pathways under their favorable pH conditions, and that are also characteristic of winter-time haze episodes (weakly acidic) (Cheng et al., 2016).

        Ionic strength impacts on $k_{chem}$ for $H_2O_2$ and $O_3$ aqueous oxidation formation pathways were included on top of the

previous modifications in the TMI_sens and TMI_NO2_sens model simulations (Chen et al., 2019; Ibusuki and Takeuchi, 1987; Maaß et al., 1999). Ionic strength adjustments were also included for S(IV) dissociation constants and Henry's law coefficients for $H_2O_2$, $O_3$ and $SO_2$ in the "All_Ionic" simulations (Ali et al., 2014; Lagrange et al., 1994; Maaß et al., 1999; Millero et al., 1989; Seinfeld and Pandis, 2016) to analyze the combined effects of ionic strength on total modelled $PM_{2.5,sulf}$ aerosol.


        In addition to the reactions shown in Table 1 (which are treated in both aerosol and cloud water), also included is S(IV) oxidation by peroxyacetic acid (PAA) and methyl hydroperoxide (MHP) in aerosol and cloud water and in-cloud S(IV) oxidation by $HNO_4$, OH and $NO_3$ (Lee and Schwartz, 1983a; Lind et al., 1987; Martin, 1984). With the exception of the "Base" case which used CMAQ's default "AQCHEM" cloud chemistry, all other simulations used the KMT2 cloud

chemistry scheme. KMT2 includes additional inorganic and organic chemistry compared to AQCHEM, including several additional S(IV) oxidation reactions as well as HMS formation and loss.

**2.4 Model Configuration**

Results of base and sensitivity simulations were compared for three different spatial domains: Fairbanks, Alaska, the

contiguous U.S (CONUS), and the Northern Hemisphere. Model simulations over Fairbanks and North Pole, Alaska, span two wintertime PM episodes (Episode 1 (E1): Jan 25th-Feb 11th, 2008 and Episode 2 (E2): Nov 4th-Nov 17th, 2008) with two days of spin-up and a horizontal resolution of 1.33 km. Model simulations over the Northern Hemispheric domain were





performed for the winter season from Dec 2015 - Feb 2016 with 2 months of spin-up and a horizontal resolution of 108 km following a standard US EPA configuration described by (Mathur et al., 2017) and (Appel et al., 2021). Model simulations
over the CONUS domain were run for the months of January and July 2016 with 10 days of spin up and at a horizontal resolution of 12km.

The Weather Research and Forecasting model (WRF (Skamarock et al., 2008)) was used to develop meteorology on all three domains. The Fairbanks WRF case follows the original configuration and case study of Gaudet et al. (2010, 2012) for
Fairbanks, AK. This older WRF simulation was updated from WRFv3.3 (Gaudet, 2010, 2012) to WRFv4.1.1. All geophysical and meteorological inputs were reprocessed for compatibility with the more recent version of WRF. Sensitivity testing found some performance improvements were realized by updating the model version including the planetary boundary layer model change from the Mellor-Yamada-Janjić (MYJ) scheme to the Mellor-Yamada-Nakanishi-Niino 2.5 order closure scheme (MYNN2.5 (Nakanishi and Niino, 2009)). Evaluations showed that the WRFv4.1.1 configuration
captured the extreme temperature variations in these cases well, with 2-m temperature root mean square error (RMSE) on the order of 2-3 K, which is within historical benchmarks for complex geographical areas that are more difficult to model (Kemball-Cook et al., 2005). For the CONUS, meteorological inputs were sourced from WRFv4.1.1 and CMAQ-inputs were processed with the Meteorology-Chemistry interface processor (MCIP (Appel et al., 2017; Otte and Pleim, 2010)) version 5.0.


The emissions inputs for the two Fairbanks wintertime PM episodes were based on inputs provided by the Alaska Department of Environmental Conservation (ADEC). These emission estimates were from the base year (2008) used for the Fairbanks $PM_{2.5}$ Moderate State Implementation Plan (SIP) and represent the best available emission estimates for the two time periods. Table 5.6-3 from section 5.06 of (ADEC, 2014) provides a summary of the methods and inputs used to develop
this emission inventory. For this model set up, we used the same inventory inputs and scripts and only updated the speciation for CMAQv5.3.2. The emission inventories for the Northern Hemispheric domain follow the same setup as described in (Appel et al., 2021) where anthropogenic emissions were sourced from the 2010 Hemispheric Transport for Air Pollution version 2 (Janssens-Maenhout et al., 2015), biogenic emissions were sourced from the Model of Emissions of Gases and Aerosols from Nature (Guenther et al., 2012), soil and lightning NO were sourced from the Global Emissions Initiative
(http://www.geiacenter.org), biomass burning emissions were sourced from the Fire Inventory from National Center for Atmospheric Research (Wiedinmyer et al., 2011), and onroad and nonroad emissions were developed using the Motor Vehicle Emission Simulator v2014a. The emission inventories for the CONUS domain were sourced from the 2016 modeling platform 2016v7.2 (beta and Regional Haze) Platform (which is documented at https://views.cira.colostate.edu//wiki/wiki/10197 and used in (Appel et al., 2021).






Gas-phase chemistry was simulated using the CB6r3 mechanism (Luecken et al., 2019) and aerosol dynamics were simulated using the aero7 module. The sulfur tracking method (STM) (which is documented at https://github.com/USEPA/CMAQ/blob/main/DOCS/Users_Guide/CMAQ_UG_ch12_sulfur_tracking.md and used in (Fahey and Roselle, 2019)) was extended to include the new heterogeneous sulfur chemical pathways in order to track the

contributions of each chemical reaction, primary emissions, and initial and boundary conditions to modelled $SO_4^{2-}$ (Appel et al., 2021).

## 2.5 Sulfate and PM$_{2.5,sulf}$ Observations

The model predictions were evaluated against available observations. While most monitoring networks report measurements

for PM$_{2.5}$ or PM$_1$ $SO_4^{2-}$, recent studies have indicated hydroxymethanesulfonate may be included in those observations (Dovrou et al., 2019; Ma et al., 2020; Moch et al., 2018; Song et al., 2019). Based on these findings, we compare measured $SO_4^{2-}$ to modelled PM$_{2.5,sulf}$:

$$PM_{2.5,sulf} \left(\frac{\mu g}{m^3}\right) = SO_4^{2-} + HMS \times \frac{MW_{SO_4^{2-}}}{MW_{HMS}}$$    (Eq. 4)

PM$_{2.5}$ $SO_4^{2-}$ observations in Fairbanks during 2008 were obtained from ADEC's Therma Electron Partisol 2000, single channel instrument w/SCC monitor 020900010 (State Office Building in Fairbanks Alaska; 64.840672, -147.722461)

(ADEC, 2023). $SO_4^{2-}$ observations for the 2016 Hemispheric Domain were sourced from the United States Environmental Protection Agency (USEPA) Air Quality System (AQS) monitoring network, the Canadian National Air Pollution Surveillance (NAP) monitoring network, the European Monitoring and Evaluation Programme (EMEP) monitoring network, and one monitor at Tsinghua University in Beijing, China (ECCC, 2022; Ma et al., 2020; Tørseth et al., 2012; USEPA). Modelled PM$_{2.5,sulf}$ concentrations and measured $SO_4^{2-}$ from the AQS and NAP networks were cast in units of micrograms

sulfur per meter cubed (µgS/m$^3$) to match the measurement units used in the EMEP and Tsinghua University $SO_4^{2-}$ measurements.

## 3 Results

### 3.1 Modelled particulate sulfur enhancement during dark and cold PM episodes in Fairbanks and North Pole, AK

#### 3.1.1 Episode 1 (E1) January 25 – February 11, 2008

The Base simulation average E1 sulfate concentrations around Fairbanks and North Pole, AK are ~ 2 - 3.5 mg/m$^3$ (Fig. 1a and c). Compared to the Base, the Base_Het simulation leads to increased $PM_{2.5,sulf}$ predictions concentrated around the cities of Fairbanks and North Pole as well as the region south of the Tanana River (Figure 1b, d). The additional multiphase chemistry in the Base_Het simulation contributes up to an additional 11 µg/m$^3$ of maximum daily $PM_{2.5,sulf}$ compared to the Base simulation in the region south of the Tanana River. Maximum daily differences are defined as:

$$Maximum\ Daily\ Differences = \max(Avg_{daily,2} - Avg_{daily,1}) \quad \text{(Eq. 5)}$$

Enhancements in $PM_{2.5,sulf}$ concentrations for the Base_Het simulation are mainly driven by increases in $SO_4^{2-}$ concentrations (increasing up to 10.9 µg/m$^3$ for daily maximum differences across the entire domain), with lesser impacts from HMS (increasing up to ~1 µg/m$^3$ for daily maximum differences across the entire domain). HMS concentrations are enhanced more in North Pole than Fairbanks, coinciding with higher HCHO emissions (Fig. S1) from residential wood combustion combined with high co-located $SO_2$ emissions (from home heating oil) along with lower temperatures (ADEC, 2017).





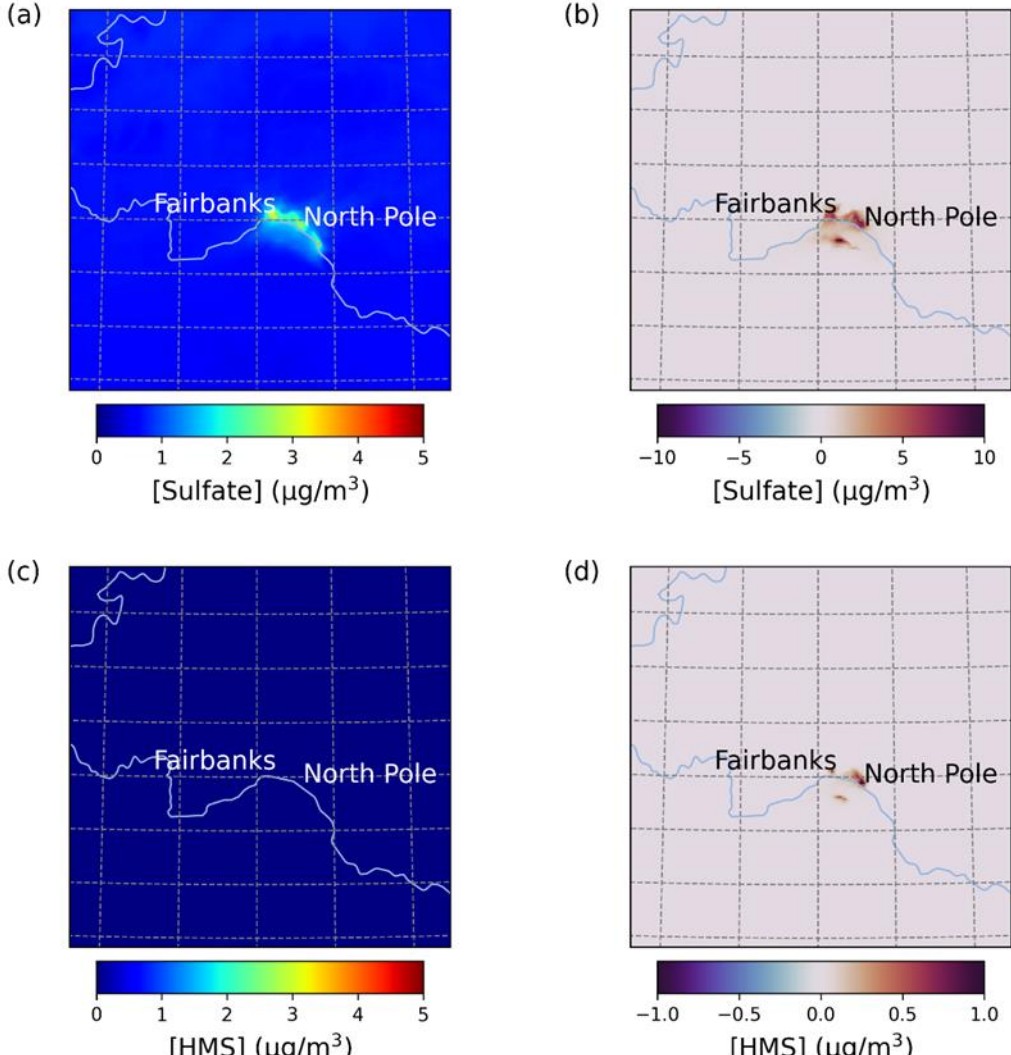

**Figure 1: Episode average sulfate (a), and HMS (c) concentrations in the Base simulation along with daily max differences in sulfate (b), and HMS (d) concentrations between the Base_Het and Base CMAQ simulations over Fairbanks and North Pole AK for episode 1 (from January 25th, 2008, to February 11th, 2008). HMS formation was not included in Base CMAQ (i.e., HMS = 0 in the Base simulation).**

Out of all of the secondary $PM_{25,sulf}$ formation pathways that are enhanced during dark cold conditions (TMI-catalyzed $O_2$, $NO_2$, and the formation of HMS), the leading secondary $SO_4^{2-}$ formation pathway in the Base_Het is the TMI-catalyzed $O_2$ oxidation pathway in ALW (Fig. 2). The first order condensed phase rate constant ($k_{chem}$) of this pathway is lower than that of the $k_{chem}$ for $NO_2$ by almost 2 orders of magnitude for average conditions characteristic of Fairbanks and North Pole for E1 (pH = 3.83, [Fe(III)] = 0.24 M, [Mn(II)] = 0.002 M, [$SO_2$] = 20 ppb, [NO2] = 20 ppb, [$SO_4^{2-}$] = 3 µg/m³, [ALW] = 6





$\mu g/m^3$, and Temp = 243K) (Fig. S2) and is ~ 1 order of magnitude higher than that for HMS formation in ALW. Despite the NO$_2$ $k_{chem}$ being higher, however, the TMI-catalyzed O$_2$ heterogeneous rate of formations rate limiting step is SO$_2$ partitioning into the particle, as Fe and Mn are both aerosol species, and simulated dark conditions reduce the conversion of

Fe$^{3+}$ to Fe$^{2+}$ from daytime photochemical reactions (Alexander et al., 2009; Rao and Collett, 1998; Shao et al., 2019). Another potential reason the TMI-catalyzed O$_2$ pathway outcompetes the NO$_2$ pathways is due to its mass accommodation coefficient ($\alpha$, Eq. 2) being higher than that for the NO$_2$ pathway by ~2 orders of magnitude. The TMI-catalyzed O$_2$ heterogeneous reactive uptake pathway also outcompetes the H$_2$O$_2$ and O$_3$ heterogeneous reactive uptake pathways due to low photochemical activity with the dark conditions of this domain and episode.


The formation of HMS is higher in North Pole, which can be colder than Fairbanks by up to ~5°C. Higher modelled HCHO emissions in North Pole (Fig. S1) along with colder temperatures, increase the partitioning of HCHO onto existing particles This effect and the increase in HMS formation is more pronounced in the TMI_sens simulation (Fig. S3 a and b).

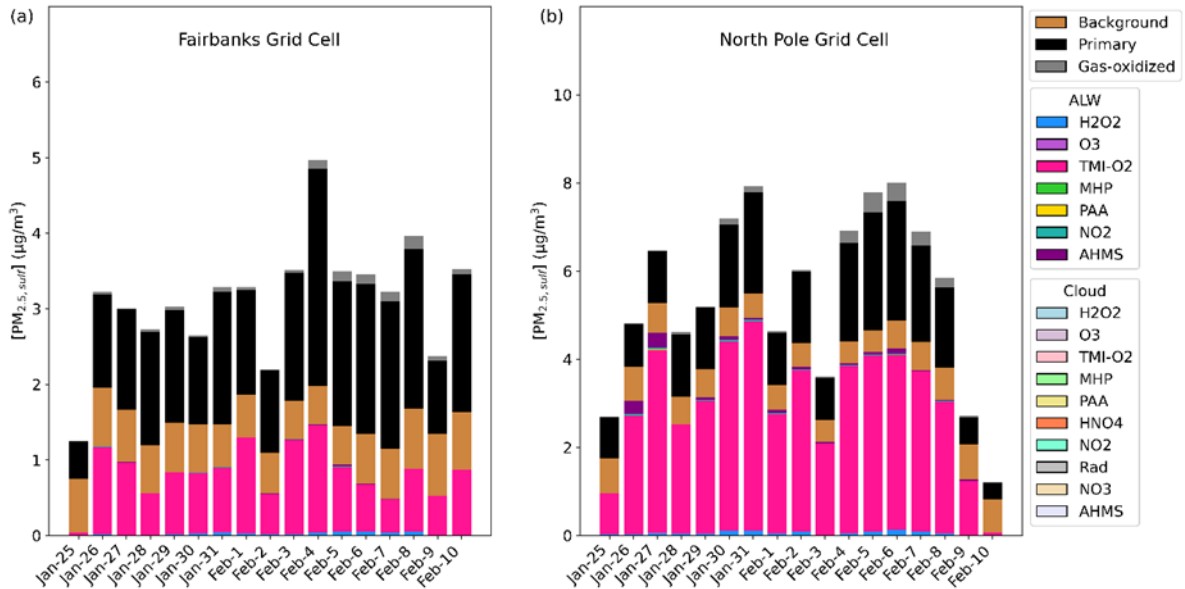


**Figure 2: Particulate sulfur process/chemistry contributions and speciation (SO$_4^{2-}$ and HMS) in Downtown Fairbanks (a) located at the State Office Building (64.84 N, -147.72 W; grid cell 108,93) and North Pole (b) located at (64.76 N, -147.34 W; grid cell 122, 86) for episode 1 (E1) speciated by source and/or formation pathway. Secondary aqueous formation of PM$_{2.5,sulf}$ is broken out into two categories: ALW and Cloud, where ALW pathways represent the heterogeneous sulfur chemistry added in this study.**


Compared to the $k_{chem}$ used in Base_Het, the alternative $k_{chem}$ in the TMI_sens for the TMI-catalyzed O$_2$ pathway is ~2 orders of magnitude lower under average conditions characteristic of this episode (Fig. S2) with the extremely cold temperatures decreasing the TMI-catalyzed O$_2$ oxidation $k_{chem}$. This ultimately results in a slower conversion of S(IV)



species to $SO_4^{2-}$ and subsequent competition with HMS formation, which increases with colder temperatures (Fig. S4).
Maximum daily average enhancements in modelled $PM_{2.5,sulf}$ concentrations in the TMI_sens simulation reach up to 25
$\mu g/m^3$ at a grid cell in North Pole and are mostly attributed to high HMS concentrations (Fig. S3 a and b). Formation rates of
HMS are also dependent on aerosol pH , which can be higher at North Pole than in Fairbanks (Fig. S4). Higher pH increases
the conversion of $HSO_3$ to $SO_3$ and the rate constant for HCHO reaction with $SO_3$ is 5 orders of magnitude higher than that
of the reaction with $HSO_3$ (Boyce and Hoffmann, 1984). Lower $SO_4^{2-}$ production rates in TMI_sens and lower modelled
aerosol acidity compared to Base_Het likely contributes to the higher HMS formation (and loss) rates seen in the TMI_sens
simulation. It is important to note that while aerosol acidity is modified by aqueous-phase formation of $SO_4^{2-}$, it is not
modified by the formation of HMS in CMAQ.

When implementing an ionic strength dependent $NO_2$ rate expression (Chen et al., 2019) on top of this alternative TMI-
catalyzed $O_2$ rate expression in the "TMI_NO2_sens" model simulation, the $NO_2$ oxidation pathway outcompeted the
formation of HMS (Fig. S3). While this model simulation compared well with the Base_Het model simulation in daily
averaged $PM_{2.5,sulf}$ concentrations, it predicted significantly lower $PM_{2.5,sulf}$ concentrations than the TMI_sens in North Pole
due to a reduction in HMS predicted. Compared to downtown Fairbanks, North Pole has lower $NO_2$ emissions but the switch
to the ionic-strength dependent reaction rate still leads to higher $SO_4^{2-}$ production in North Pole compared to TMI_sens.


The "ALL_Ionic" model simulation resulted in similar $PM_{2.5,sulf}$ predictions and pathway contributions as the
TMI_NO2_sens simulation for this episode (Fig. S3). In both Fairbanks and North Pole, $SO_4^{2-}$ formation attributed to
heterogeneous reactive uptake via the $H_2O_2$ oxidation pathway increased slightly. For ionic strength factor calculations, ionic
strength is capped at the maximum ionic strength considered in the experiments the parameterizations are based on (~5-6M)
(Ali et al., 2014; Millero et al., 1989). The modelled ionic strength of ALW is typically at or above the maximum
experimental ionic strength when deriving both the effective Henry's law coefficients, dissociation coefficients, and $k_{chem}$
for the $H_2O_2$ oxidation pathway – leading to expected higher dissolution and kinetics for this pathway. HMS concentrations
for this model simulation also increased slightly in comparison to the TMI_NO2_sens. Assuming excess of dark oxidant pre-
cursors (TMI, $NO_2$ and HCHO) and maximum ionic strength, the leading higher HMS production rate in the ALL_Ionic is
likely due to a slightly higher range of pH (~2-5.5) compared to the Base_Het and TMI_NO2_sens simulations (Fig. S3d).

### 3.1.2 Episode 2 (E2) November 4 – 17, 2008

Sulfate and HMS are known to form efficiently in cloud and fog droplets (Altwicker and Nass, 1983; Boyce and Hoffmann,
1984; Calvert et al., 1978; Clifton et al., 1988; Ibusuki and Takeuchi, 1987; Lee and Schwartz, 1983a; Martin and Good,
1991; McArdle and Hoffmann, 1983). In E1, there was minimal cloud or fog liquid water simulated; however, during E2





(November 4 – November 11, 2008), there were some periods where cloud/fog chemistry impacts on $PM_{25,sulf}$ formation were evident.

Compared to E1, $PM_{2.5,sulf}$ concentration enhancements were lower overall during E2. Differences between Base_Het and

Base simulations, however, are appreciable during this episode with $PM_{2.5,sulf}$ increasing up to 4.6 µg/m$^3$ across the entire domain (daily maximum difference) (Fig. 3). Enhancements in $PM_{2.5,sulf}$ are mainly driven by increased $SO_4^{2-}$ formation in and around Fairbanks and North Pole; however, simulated HMS concentrations reached up to 4.4 µg/m$^3$ south of the Tanana River (daily maximum) for this episode. Note that the Base simulation included some contributions from in-cloud S(IV) oxidation (i.e., 5 S(IV) oxidation reactions from CMAQ's default cloud chemistry mechanism (AQCHEM)), while Base_Het

includes the additional in-cloud chemical reactions from the KMT2 cloud chemistry module.



**Figure 3. Episode average sulfate (a), and HMS (c) concentrations in the Base simulation along with daily max differences in sulfate (b), and HMS (d) concentrations between the Base_Het and Base CMAQ simulations over Fairbanks and North Pole AK for episode 2 (from November 4th, 2008, to November 17th, 2008). HMS formation was not included in Base CMAQ (i.e., HMS = 0 in the Base simulation).**

In the Base_Het simulation of Episode 2, $PM_{2.5,sulf}$ was formed both in ALW and cloud liquid water (Fig. 4). Similar to Episode 1, the leading secondary formation pathway in downtown Fairbanks was the TMI-catalyzed $O_2$ pathway in both ALW and cloud water. This formation pathway split between the ALW and cloud formation pathways when modelled fog water content was around 0.025-0.05 g/m³ for Nov. 5th and Nov. 7th, however, was completely overtaken by the cloud



formation pathway on Nov. 15[th] when fog water content was > 0.175 g/m$^3$ (at both Fairbanks and North Pole), highlighting that when surface clouds, or fog water, is present, reactions in cloud water can compete with those in aerosol water. In North Pole, the leading ALW PM$_{2.5,sulf}$ formation pathway was also the TMI-catalyzed O$_2$ oxidation pathway with contributions

from this pathway in cloud water as well. Sulfate formed via S(IV) oxidation by NO$_2$ in cloud water was slightly higher in North Pole than downtown Fairbanks for all cloud events even though NO$_2$ emissions are higher in downtown Fairbanks (Fig. S5). This could be due to higher cloud/fog pH in North Pole compared to downtown Fairbanks, as this pathway is known to favor pH > 5 (Clifton et al., 1988; Lee and Schwartz, 1983a; Littlejohn et al., 1993; Sarwar et al., 2013). HMS formation in ALW is also present in North Pole on the last day of Episode 2, corresponding with temperatures around -28 ℃

and aerosol pH ~5.

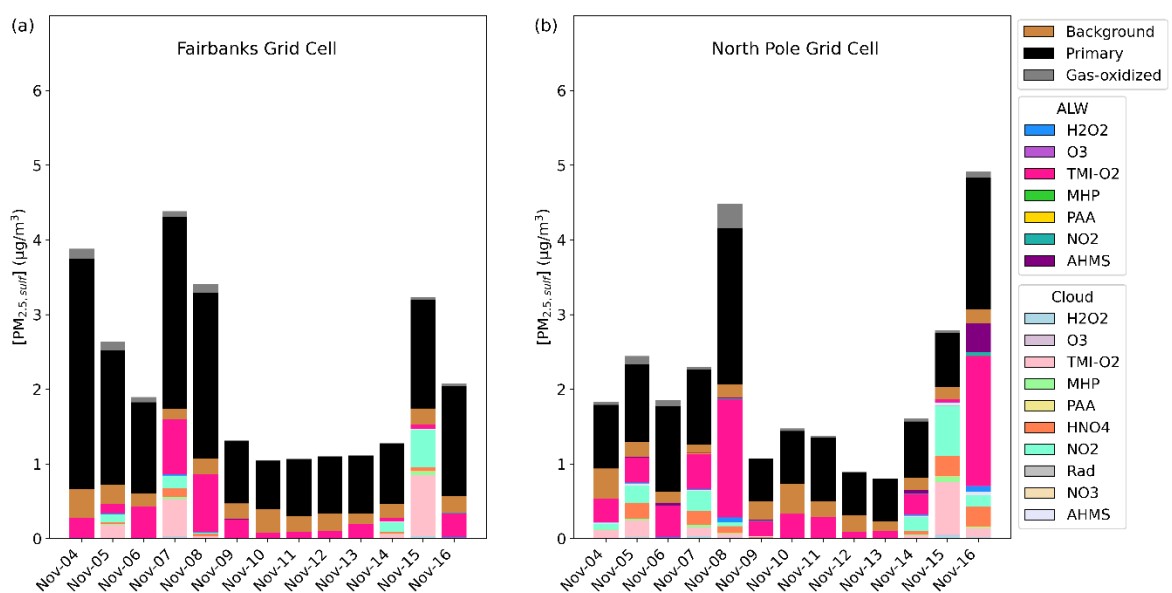

**Figure 4. Particulate sulfur process/chemistry contributions and speciation (SO42- and HMS) in Downtown Fairbanks (a) located at the State Office Building (64.84 N, -147.72 W; grid cell 108,93) and North Pole (b) located at (64.76 N, -147.34 W; grid cell 122,**
**86) for episode 2 speciated by source and/or formation pathway. Secondary aqueous formation of PM2.5,sulf is broken out into two categories: ALW and Cloud, where ALW pathways represent the heterogeneous sulfur chemistry added in this study.**

HMS contribution to PM$_{2.5,sulf}$ concentrations is further increased on Nov. 16[th] in North Pole in the TMI_sens model simulation (Fig. S7b). Despite this increase in HMS, total PM$_{2.5,sulf}$ predicted in this simulation is lower than in the Base_Het

simulation for most of the episode. The compensation of the HMS formation pathway for the TMI-catalyzed O$_2$ formation pathway is not as significant for this episode despite similar HCHO emissions (Fig. S5) and is likely due to higher temperatures (~ +15 °C warmer than E1).



### 3.1.3. Improved model performance for PM$_{2.5,sulf}$ in Fairbanks

Daily average PM$_{2.5,sulf}$ concentrations for the Base and Base_Het simulations were compared with 24-hour SO$_4^{2-}$
measurements taken every third day at the State Office Building in downtown Fairbanks (Fig. 5) (ADEC, 2023).

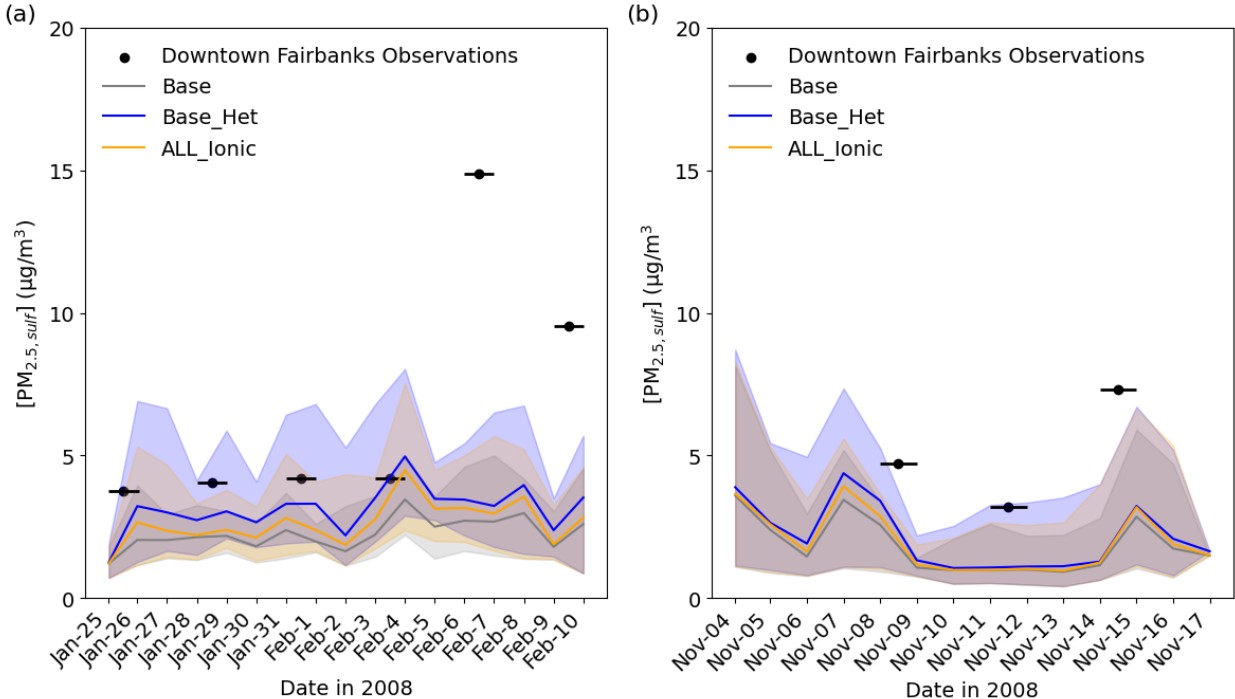

**Figure 5. Timeseries comparing modelled PM$_{2.5,sulf}$ concentrations to measured SO$_4^{2-}$ concentrations at the State Office Building in Downtown Fairbanks (64.84 N, -147.72 W; grid cell 108,93) for episode 1 (a) and episode 2 (b). Lines represent daily average**
**modelled values and shading represent hourly maximum and minimum concentrations for each day. Black lines represent the sampling period of the monitor measurements with black circles representing the mid-point of the 24-hour sampling period.**

With the inclusion of heterogeneous sulfur chemistry in CMAQ (Base_Het), the mean bias in PM$_{2.5,sulf}$ improved by ~0.62
μg/m$^3$ during E1 and by ~0.36 μg/m$^3$ during E2 , reducing the model bias by up to ~1 μg/m$^3$ during E1 and up to ~ 0.85
μg/m$^3$ during E2. CMAQ still underpredicts PM$_{2.5,sulf}$ observations for both episodes, particularly during E1 for February 6$^{th}$
and 9$^{th}$ by ~11 and 7 μg/m$^3$ respectively. PM$_{2.5}$ speciation measurements during both episodes were only available for a
single monitor located in Downtown Fairbanks, and therefore we were not able to assess model performance in North Pole,
where modelled PM$_{2.5,sulf}$ concentrations can be higher by up to ~5-27 μg/m$^3$ (daily maximum difference) for E1 and ~2-2.4
μg/m$^3$ (daily maximum difference) for E2. Although there were no measurements to compare with modelled North Pole
results, higher PM$_{2.5,sulf}$ enhancements at North Pole are consistent with higher PM$_{2.5}$ concentrations at North Pole compared
to Fairbanks (ADEC, 2017).





**3.2 Modelled PM$_{2.5,\text{sulf}}$ formation enhancements over the Northern Hemisphere in Winter**

To investigate model performance with the addition of heterogeneous sulfur chemistry for other locations and time periods that experience dark and cold conditions, Hemispheric CMAQ (HCMAQ) was run over the Northern Hemisphere from

October 2015 to February 2016 (with the first two months as model spin up) for the same base and sensitivity simulations as above. PM$_{25,\text{sulf}}$ in China was of interest on this domain, given its haze events. High secondary and heterogeneously formed SO$_4^{2-}$ and HMS have been documented to coincide with the high SO$_2$ emissions, high PM loadings, and high ALW during Chinese haze events (Cheng et al., 2016; Elser et al., 2016; Li et al., 2017; Ma et al., 2020; Peng et al., 2021; Wang et al., 2016). Winter episode average modelled SO$_2$ over the northeast China and other parts of China can be > 20 ppb, episode

average modelled PM$_{2.5}$ can be > 80 µg/m$^3$, and episode averaged ALW can be > 80 µg/m$^3$ in this region (Fig. S8).

Maximum enhancements in PM$_{2.5,\text{sulf}}$ for the Base_Het simulation occur largely in the North China Plain and northeast China with some notable enhancement over India as well (Fig. 6) and led to a maximum daily increase in PM$_{2.5,\text{sulf}}$ by up to ~54 µgS/m$^3$ at a grid cell in southern China (27.1651 N, 107.5234 W; grid cell [149,67]) for this winter period (Fig. 6).





**Figure 6. Episode average sulfate (a), and HMS (c) concentrations in the Base simulation along with maximum daily differences in sulfate (b), and HMS (d) concentrations between the Base_Het and Base CMAQ simulations over the Northern Hemisphere for a winter-time episode (from December 1st, 2015, to February 29th, 2016). HMS formation was not included in Base CMAQ. Differences are cast in micrograms of sulfur per meter cubed to be consistent with measurement units from EMEP.**

This maximum daily enhancement is due almost entirely to an increase in predicted $SO_4^{2-}$ (max daily concentration of 53 $\mu gS/m^3$ at this same location). On average $SO_4^{2-}$ can increase by up to ~9 $\mu gS/m^3$ over a grid cell in northeastern China



(45.6698 N, 127.9877 W; grid cell [122,64]). HMS contributions to $PM_{2.5,sulf}$ for the Base_Het run over this domain were less significant with a maximum daily concentration of ~2.6 µgS/m$^3$ in a grid cell near Tehran, Iran (36.7976 N, 51.6208 W;

grid cell [137,119]).

For the TMI_sens simulation, maximum daily $PM_{2.5,sulf}$ concentrations increased from the Base simulation by up to ~33 µgS/m$^3$ and on average up to ~7 µgS/m$^3$ at a grid cell in Hebei, China (39.8603 N, 119.2348 W; grid cell [131,65]) (a reduced episodic enhancement in comparison to the Base_Het simulation). This maximum enhancement is also almost

entirely attributed to $SO_4^{2-}$ increases. HMS concentrations in the TMI_sens simulation, however, contributed up to ~2.8 µgS/m$^3$ (in maximum daily concentrations) in a grid cell in northeast China (45.6698 N, 127.9877 W; grid cell [122,64]) (Fig. S9). Predicted $PM_{2.5,sulf}$ concentrations in the TMI_NO2_sens and All_Ionic simulations were similar to the TMI_sens simulation for both predicted $SO_4^{2-}$ and HMS concentrations with the exception of not reproducing as high of HMS concentrations in northeast China (Fig. S9).


The addition of heterogenous sulfur chemistry decreased the model bias during an extreme haze event on the HCMAQ domain as well. Modelled $PM_{2.5,sulf}$ concentrations from a grid-cell over Beijing were compared to sulfate measurements at Tsinghua University in Beijing (Zhang et al., 2021b) (Fig. 7).





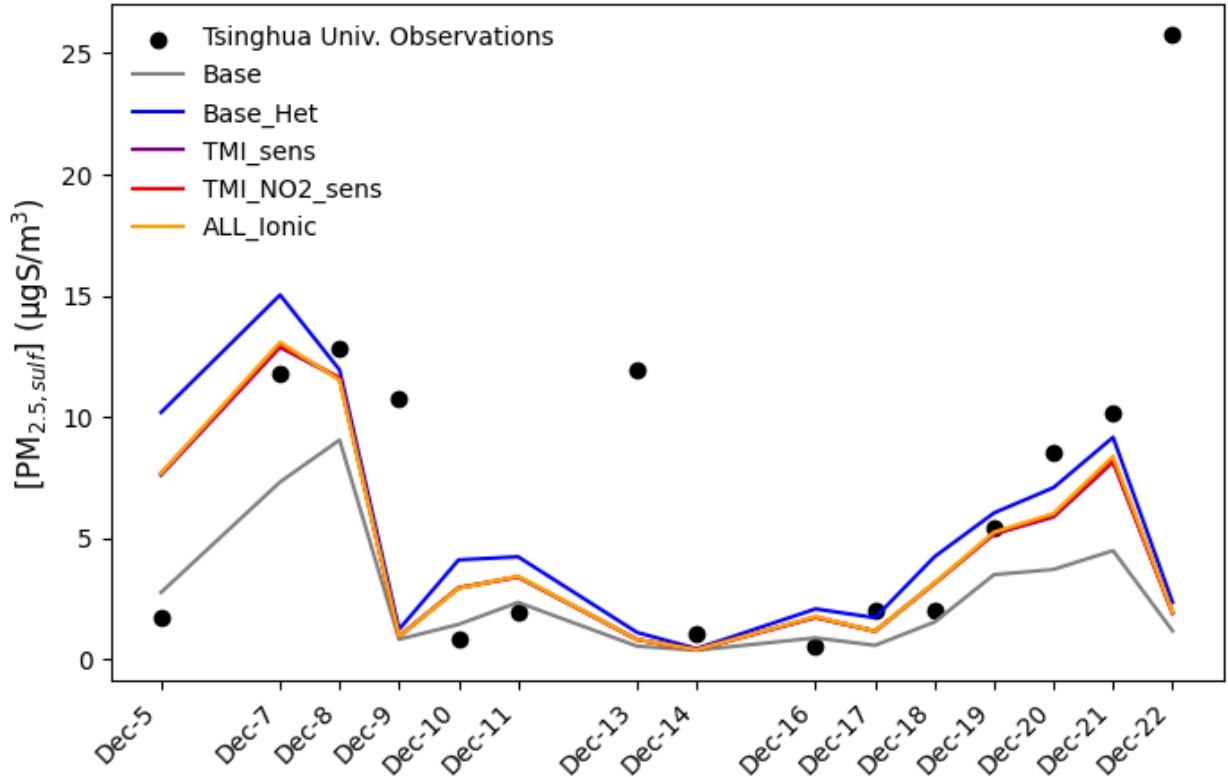


**Figure 7. Model-measurement comparisons of PM$_{2.5,sulf}$ across all model runs for HCMAQ at a grid cell over Tsinghua University in Beijing, China from December 5$^{th}$ to December 22$^{nd}$, 2015. Observations sourced from Zhang et al. (2021b)**

The Base_Het and all additional sensitivity runs predicted higher PM$_{2.5,sulf}$ at this grid cell than the Base model simulation

and reduced modelled mean bias by 2.7 µgS/m$^3$ (model mean bias with Base was -4.5 µgS/m$^3$ and mean bias with Base_Het was -1.8 µgS/m$^3$) (Fig. 7). This enhancement in PM$_{2.5,sulf}$ is mainly attributed to SO$_4^{2-}$ increases, as HMS maximum daily concentrations over this grid cell for this time period were ~ 0.002 ugS/m$^3$ for Base_Het and the three sensitivity simulations. Simulated HMS for Base_Het and the sensitivities for the winter 2016 case were also ~0.002 ugS/m$^3$ (~0.006 ug/m$^3$). While the HCMAQ estimates may be low, they are within the range of daily filter measurements of a polluted episode (from ~ 0 to

1.2 µg/m$^3$) reported in Ma et al. (2020).

Normalized mean biases (NMB) for PM$_{2.5,sulf}$ in the Base simulations over the N. Hemisphere ranged between -90% to 800% and NMB for the Base_Het simulation ranged from -90% to 980% (Fig. S10). The largest positive NMB occurred at a site in the Western U.S. for both simulations and is due, however, to very low modelled and observed concentrations. The mean

biases at this location was only ~0.1-0.2 ugS/m$^3$ for both simulations, as this region was generally not affected by our updates (Fig. 6). Improvement of negative NMB occurred in the Base_Het run in some parts of the eastern U.S. and Canada,



however caused and/or increased NMB in the positive direction in most of the eastern U.S. and Canada (Fig. S10). The change in NMB in Europe was not significant, despite enhancements from HMS formation. Regional aggregated model performance metrics can be found in Table S1.

**3.3 CMAQ model performance changes over the Contiguous United States with the implementation of heterogeneous sulfur chemistry**

CMAQ was run over the CONUS domain for both a winter and summertime episode to ensure that heterogeneous sulfur chemistry updates intended to be significant during dark and cold episodes had minimal impact on domains and episodes where this chemistry is less likely to dominate. For a January 2016 simulation over the CONUS domain $PM_{2.5,sulf}$

enhancements can be seen mainly over the eastern U.S. and western Canada (Fig. 8), on average were enhanced by up to ~ 1.5 µg/m$^3$. Average daily enhancements in Base_Het $SO_4^{2-}$ for the entire episode were up to ~1.5 µg/m$^3$ as well and daily average Base_Het HMS were up to ~0.7 µg/m$^3$.

$PM_{2.5,sulf}$ maximum daily enhancements for this episode, however, reached 28 µg/m$^3$ at a grid cell in southwestern Kansas
(37.3983 N, -101.9184 W; grid cell [121,177]). It should be noted that total modelled daily averaged $PM_{2.5}$ concentrations at this grid cell was 575 µg/m$^3$ signifying a major PM event. $PM_{2.5,sulf}$ enhancements from $SO_4^{2-}$ at this grid cell and day contributed ~75% with HMS contributing ~ 25%. The leading $PM_{2.5,sulf}$ formation pathway at this grid cell, however, in the Base_Het run was the TMI-catalyzed $O_2$ oxidation pathway in ALW (Fig. S11). This pathway dominates at a few other locations (Fig. S11), however, gas-phase oxidation of $SO_2$ by OH is the leading secondary $PM_{2.5,sulf}$ formation pathway
spatially, followed by cloud-aqueous oxidation by $H_2O_2$ and $HNO_4$.

The maximum daily average enhancement in HMS was ~ 13 µg/m$^3$ and occurred in the Ozarks in south central Missouri (36.6721 N, -92.9428 W; grid cell [114,243]) also coinciding with a major PM event (model daily average $PM_{2.5}$ concentrations of 301 µg/m$^3$) (Fig. 8).




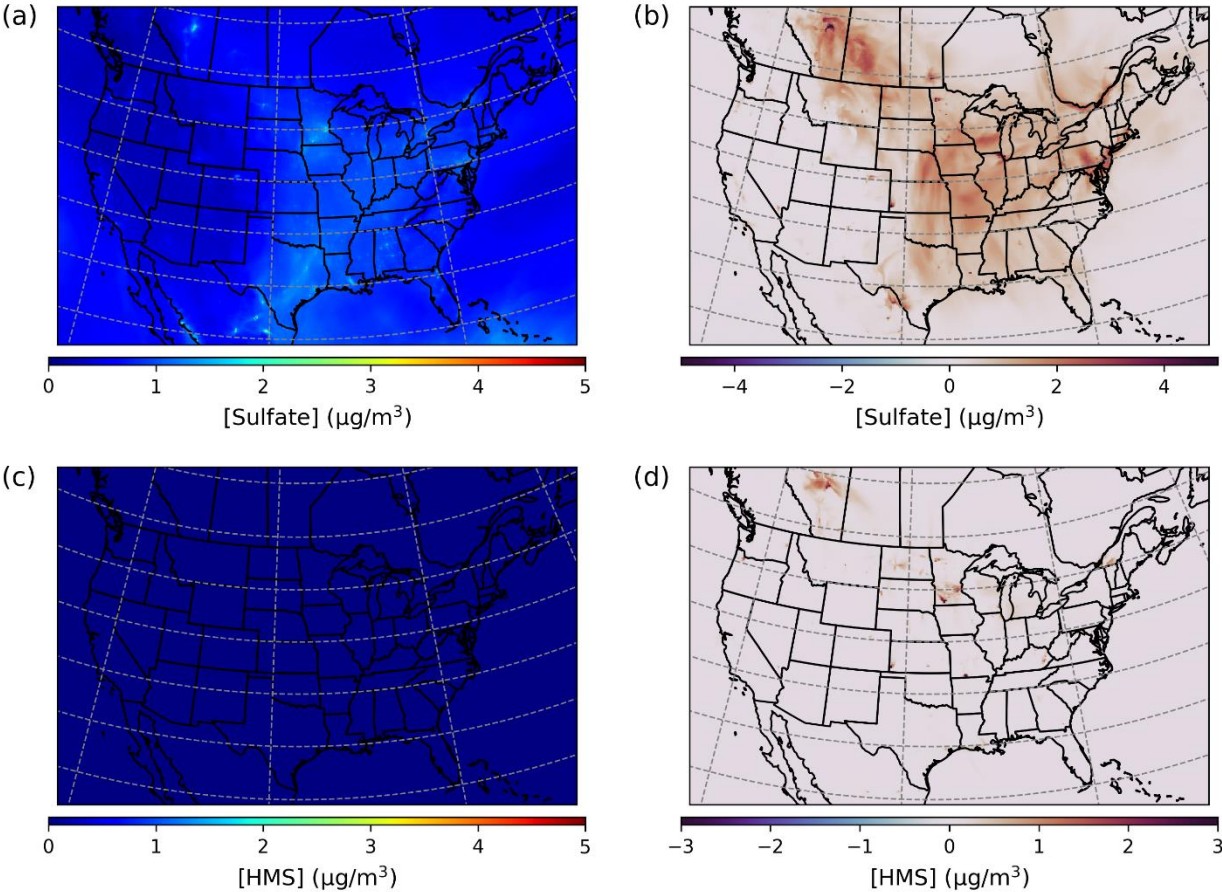

**Figure 8. Episode average sulfate (a), and HMS (c) concentrations in the Base simulation along with maximum daily differences in sulfate (b), and HMS (d) concentrations between the Base_Het and Base CMAQ simulations over the Contiguous United States for**
**a winter-time episode (January 2016). HMS formation was not included in Base CMAQ, and HMS mass concentrations are multiplied by the ratio of sulfate to HMS molecular mass.**

$PM_{2.5,sulf}$ concentration enhancements in the TMI_sens simulation are similar to those in Base_Het with a lesser contribution from $SO_4^{2-}$ production and a higher contribution from HMS production (Fig. S12 a and b). The highest maximum daily

enhancement for the TMI_sens simulation occurred in the same grid cell in southwest Kansas discussed previously and was slightly lower than the Base_Het simulations maximum daily average enhancement (27.7 µg/m³). The percentage of $PM_{2.5,sulf}$ that was $SO_4^{2-}$ and HMS, however, was ~10% (3.2 µg/m³) and ~90% (28.3 µg/m³ or 24.5 µg/m³-$SO_{4,eq}$). Other instances of HMS concentrations higher than 5 µg/m³ for this run were infrequent and either coincided with major $PM_{2.5}$ events ($PM_{2.5}$ concentrations > 100 µg/m³). The highest maximum daily enhancement in $PM_{2.5,sulf}$ for the TMI_NO2_sens and All_Ionic

runs were 22 and 18 µg/m³ respectively and occurred at the same grid cell in southwestern Kansas. Enhancements in $PM_{2.5,sulf}$ in this grid cell were mainly attributed to HMS for both the TMI_NO2_sens and All_Ionic simulations and were





~14-15 µg/m³, further demonstrating the importance of HMS to total PM$_{2.5,sulf}$ when a temperature and pH dependent TMI-catalyzed O$_2$ $k_{chem}$ is used during a cold major PM event.

Both the Base and Base_Het CONUS simulations overestimate measurements in the western U.S. (Fig. 9) with the highest positive NMB ~450% in Washington state with little difference in the NMB between the two runs – as this region was generally not affected by the implementation of heterogeneous sulfur chemistry (Fig. 8). It should also be noted that mean bias in the western U.S. is fairly low (Fig. S13), and therefore the high NMB in this region is a result of overall low PM$_{2.5,sulf}$ concentrations (both modelled and measured). Negative NMB of PM$_{2.5,sulf}$ in the eastern U.S. in the Base run are ameliorated

in the Base_Het runs, particularly in the Ohio River Valley and southeast U.S.; however, areas in the east with good model PM$_{2.5,sulf}$ performance in the Base run were mostly overpredicted in the Base_Het run. Model performance differences across the additional sensitivity runs were minimal however all sensitivity runs had a lower positive NMB at the AQS sites around Chicago compared to the Base_Het (Fig. S14), and the TMI_sens simulation had the lowest positive NMB and mean bias at the AQS site in south central Missouri out of all of the newly implemented model runs.


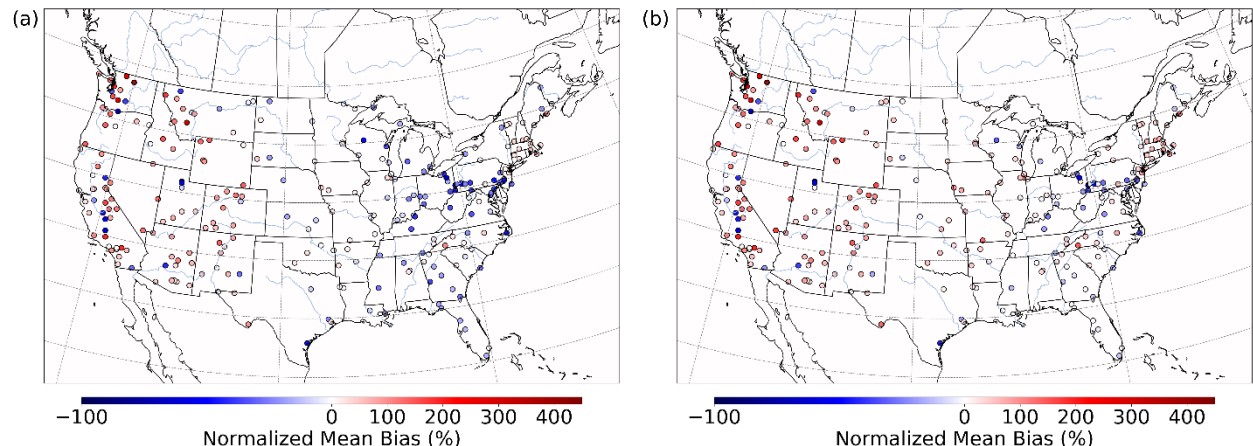

**Figure 9. Normalized Mean Bias in PM2.5,sulf concentrations by monitor over the CONUS domain for a winter-time episode (January 2016) for Base CMAQ (a) and Base_Het CMAQ (b).**

For a July 2016 simulated episode over the CONUS domain, overall daily enhancements in Base_Het PM$_{2.5,sulf}$ were on average 0.05 µg/m³ across the entire domain. Daily SO$_4^{2-}$ enhancements were more prevalent spatially over the eastern part of the U.S., particularly in the South and Ohio River Valley (Fig. 10), and HMS formation was more prevalent in the western part of the domain.

Maximum daily PM$_{2.5,sulf}$ enhancements in the Base_Het model run reached up to 789 µg/m³, at a grid cell in Monterey

County, California (36.2839 N, -121.9208 W; grid cell [135,30]) on July 26[th] largely due to high daily averaged HMS concentrations (~533 µg/m³) (Figure 10) and coincide with the Soberanes fire at this location in late July (Queally, 2016).





$SO_4^{2-}$ concentration enhancements were also significant at this grid cell (~255 µg/m$^3$). It should be noted that modelled daily averaged PM$_{2.5}$ and HCHO concentrations for this day and grid cell were also extremely high (9389 µg/m$^3$ and 577ppbV respectively).


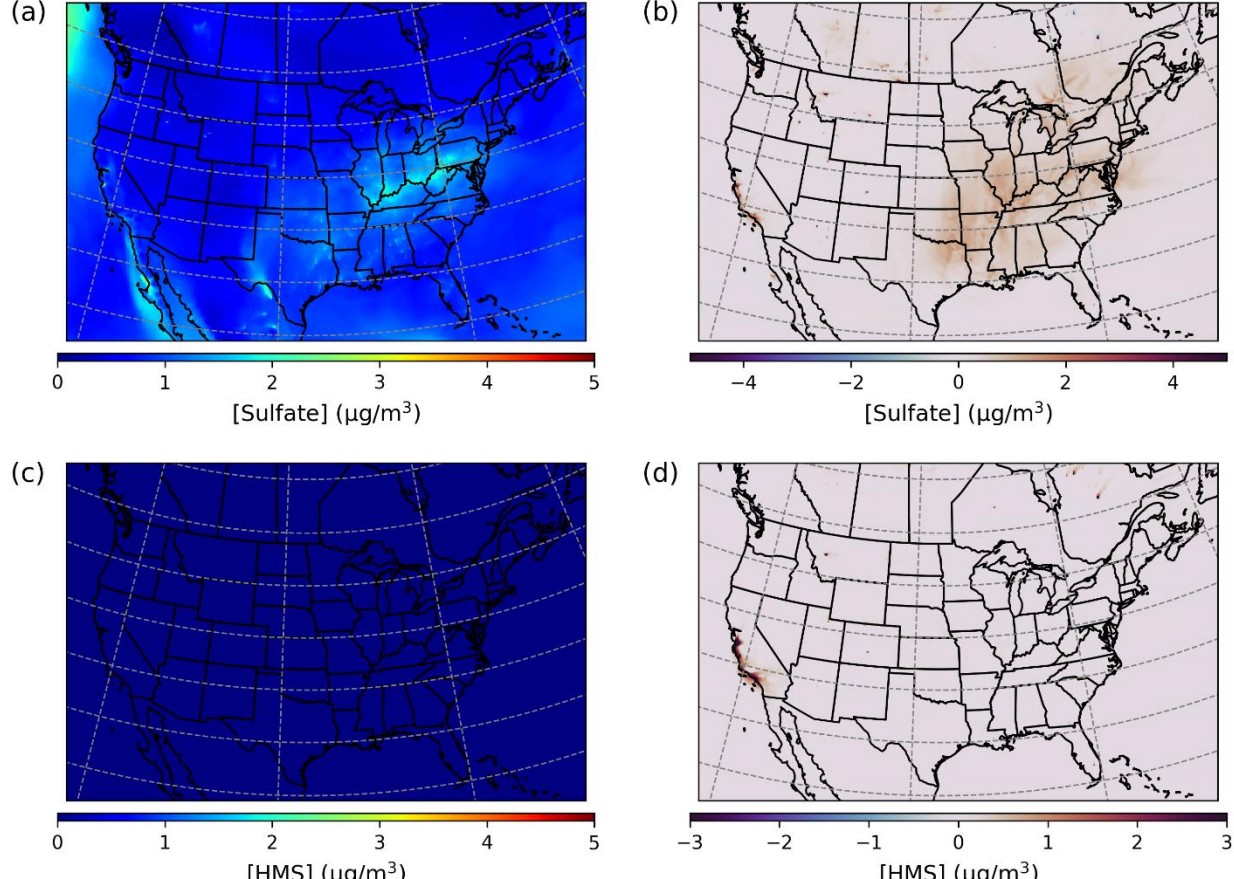

**Figure 10. Episode average sulfate (a), and HMS (c) concentrations in the Base simulation along with maximum daily differences in sulfate (b), and HMS (d) concentrations between the Base_Het and Base CMAQ simulations over the Contiguous United States for a summer-time episode (July 2016). HMS formation was not included in Base CMAQ, and HMS mass concentrations are**
**multiplied by the ratio of sulfate to HMS molecular mass.**

In the same grid cell and time where there were maximum daily enhancements in PM$_{2.5,sulf}$ in the Base_Het run occur, TMI_sens simulated PM$_{2.5,sulf}$ increased by ~791 µg/m$^3$ with HMS contributing ~732 µg/m$^3$. Similar to results in other domains and episodes, in the TMI_sens simulation, HMS formation is higher and $SO_4^{2-}$ enhancement decreases in comparison to the Base_Het. Average daily $SO_4^{2-}$ enhancements in the TMI_sens reached up to 4 µg/m$^3$ and in the Base_Het
reached up to 11 µg/m$^3$, and both occurred in the same grid cell in Monterey, California mentioned before. For the TMI_NO2_sens simulation, maximum daily enhancement in PM$_{2.5,sulf}$ concentrations are similar to the TMI_sens (~791




µg/m³) with a slightly lower contribution from HMS (~720 µg/m³). Maximum daily PM$_{2.5,sulf}$ enhancements were the lowest in the All_Ionic model run, however still a substantial enhancement (~788 µg/m³; occuring in the same grid cell and time as all other models run maximum daily enhancements) with HMS contributing ~709 µg/m³ and SO$_4^{2-}$ contributing ~ 79 µg/m³

to this enhancement. Spatial distribution of SO$_4^{2-}$ enhancements for all sensitivity runs were similar to the Base_Het SO$_4^{2-}$ enhancements (Fig. S15).

Both the Base and Base_het simulations mainly overestimate PM$_{2.5,sulf}$ concentrations in the northwest U.S. and have similar performance (Fig. 11 and Fig. S16) due to this region not generally being affected by heterogeneous sulfur chemistry updates
(with the exception of a few locations that aren't in the same grid cells as an AQS monitor). Although there were significant enhancements in HMS and off the Californian coast, HMS concentrations greater than 5 ug/m³ were limited to a few grid cells on July 26$^{th}$ (when maximum concentrations of HMS were predicted) and did not reach the nearest AQS monitors. Higher daily HMS concentrations were predicted south of Monterey down the California Coast the following 2 days (which did not have a corresponding AQS measurement). Increased SO$_4^{2-}$ formation in the Base_Het simulation improved modelled
underestimates in some parts of the Eastern U.S. (Fig. 11 and Fig. S16), particularly in parts of the southeast (Fig. S17).

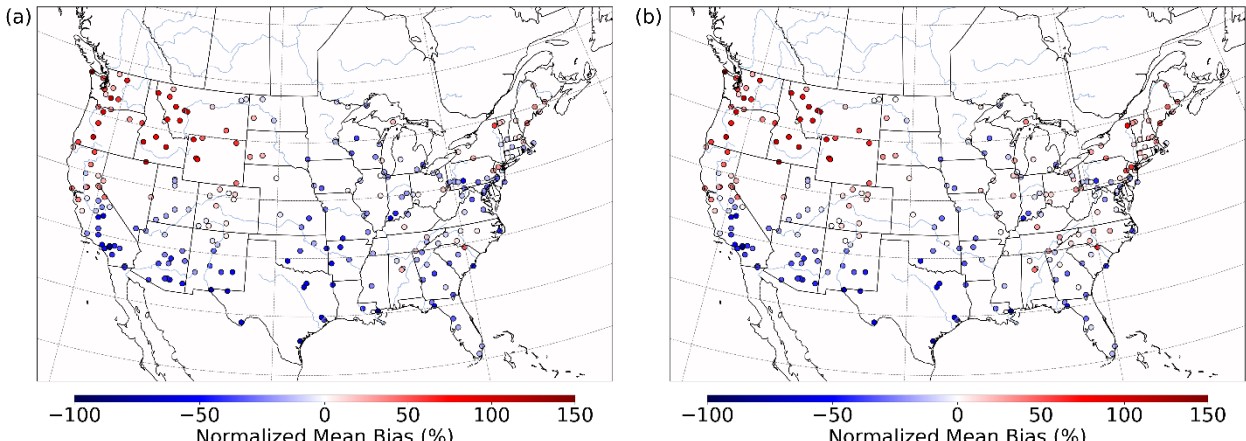

**Figure 11. Normalized Mean Bias in PM2.5,sulf concentrations by monitor over the CONUS domain for a summer-time episode (July 2016) for Base CMAQ (a) and Base_Het CMAQ (b).**

Although the implementation of heterogeneous sulfur chemistry generally led to minimal enhancements in both SO$_4^{2-}$ and HMS over the entire CONUS domain for both episodes, it did slightly increase SO$_4^{2-}$ in the eastern U.S. and substantially increased HMS concentrations during major PM events, including but not limited to wildfires. Enhancements in SO$_4^{2-}$ could potentially impact the formation of secondary organic aerosol (Fan et al., 2022). SO$_4^{2-}$ is an important nucleophile in the formation of isoprene-epoxydiol (IEPOX)-derived organosulfates (Surratt et al., 2010) and enhancements in SO$_4^{2-}$
concentration may help reduce model underpredictions for IEPOX organosulfates (Budisulistiorini et al., 2017). Aqueous



$SO_4^{2-}$ in the presence of $Fe^{3+}$ and other isoprene-derived SOA precursors has been shown to enhance the production of C2-C4 organosulfur compounds as well (Huang et al., 2019). Modelled enhancements in both $SO_4^{2-}$ and HMS during wildfires are both particularly high due to extraordinarily high emissions during these events in general. The TMI_sens predicted the highest HMS concentrations across all model simulations, however max daily $SO_4^{2-}$ concentrations in the TMI_NO2_sens

were higher in comparison to the TMI_sens, indicating the importance of the heterogeneous $NO_2$ oxidation pathway during wildfire events. Wildfires release high concentrations of black carbon which can inhibit modelled photolysis of $NO_2$, and high in-plume oxidant concentrations can facilitate it's regeneration (Buysse et al., 2019; Xing et al., 2017).

## 4 Discussion

### 4.1 Improved model performance with heterogeneous sulfur chemistry in dark cold episodes

Traditional mechanisms of secondary sulfate formation have not been able to reproduce high observed sulfate concentrations experienced during Fairbanks and North Pole, AK winters. The added heterogeneous sulfur chemistry in ALW in this study enhanced modelled wintertime $PM_{2.5,sulf}$ concentrations in this domain as well as in China, where high wintertime $PM_{2.5,sulf}$ concentrations have also been observed  (ADEC, 2017; Cheng et al., 2016; Ma et al., 2020; Moch et al., 2018; Wang et al., 2016; Zhang et al., 2021b). Inclusion of these reactions improved model performance in these regions (Fig. 5 and Fig. 7).

$PM_{2.5,sulf}$ enhancements between the Base and Base_Het simulations during a PM episode (E1) in the Fairbanks domain showed that heterogeneous sulfur chemistry in ALW can increase daily $PM_{2.5,sulf}$ concentrations by up to 11 µg/m$^3$ across the entire domain for this dark and cold modeling episode (Fig. 2). At grid cells in downtown Fairbanks and North Pole, maximum daily enhancements in $PM_{2.5,sulf}$ during E1 can range ~1-1.5 µg/m$^3$ and ~ 4-25 µg/m$^3$ respectively (Fig. 2 and S3), across all of the heterogeneous sulfur chemistry configurations tested in this work. In this same domain during E2, modelled

secondary $PM_{2.5,sulf}$ formation takes place primarily via in-cloud formation pathways (Fig. 4 and S7), highlighting the enduring importance of cloud-aqueous chemistry in modeling $PM_{2.5,sulf}$ formation when substantial liquid cloud water is present.

$PM_{2.5,sulf}$  enhancements from heterogeneous sulfur chemistry for the wintertime HCMAQ simulations occurred mostly in

China, South Asia and Europe (Fig. 6). Maximum daily enhancements in $PM_{2.5,sulf}$ ranged between 33-54 µgS/m$^3$ across heterogeneous sulfur chemistry configurations tested in this work and were all located in China. Heterogeneous sulfur chemistry updates also improved negative model bias for $PM_{2.5,sulf}$ in Canada (Table S1) where wintertime temperatures can be extremely cold and residential home heating along with industrial emissions may be high (Cho et al., 2009; Liggio et al., 2016; Stroud, 2009).


In general, there were minimal changes in predicted $PM_{2.5,sulf}$ in regions and during episodes that do not experience extreme cold and dark conditions. Positive model bias in estimating $PM_{2.5,sulf}$ in Europe for the HCMAQ simulation, slightly increases in the Base_Het, TMI_sens, and TMI_NO2_sens simulations, however, remains unchanged in the All_Ionic simulation. Similarly, model bias in estimating $PM_{2.5,sulf}$ in the U.S. for the HCMAQ also increases slightly from the Base in all the newly implemented model runs by 0.05 µg/m$^3$ (Table S1).

The inclusion of heterogeneous sulfur chemistry had a lesser impact on modelled $PM_{2.5,sulf}$ over the CONUS domain (in comparison to the Fairbanks and HCMAQ domains) with the exception of high PM events. Interestingly, the inclusion of these missing pathways reduced some of the negative bias in $PM_{2.5,sulf}$ concentrations in the Southeast during warm and light summertime conditions (Fig. 11, S16, and S17) and are mostly from $SO_4^{2-}$ enhancements (Fig. 10 and S15). $SO_4^{2-}$ can participate as a reactant and a modulator of pH in the heterogeneous formation of IEPOX derived organosulfates (Marais et al., 2016; Pye et al., 2017; Pye et al., 2020) and improved $SO_4^{2-}$ negative bias may also improve model performance of IEPOX-SOA.

## 4.2 Contribution of HMS to $PM_{2.5,sulf}$

In this study we found that HMS, a previously un-tracked aerosol species in CMAQ, can contribute substantially to total $PM_{2.5,sulf}$ dependent on HCHO and $SO_2$ concentrations, temperature, and heterogeneous sulfur chemical kinetics chosen. During E1 in Fairbanks, HMS concentrations in North Pole are higher in comparison to Fairbanks. Given stagnant conditions for this domain and episode, emissions tend to stay local (ADEC, 2017; Gaudet, 2012; Tran, 2011) . Thus, HCHO from residential wood combustion and $SO_2$ from home heating oil in North Pole, a largely residential area, compared to downtown Fairbanks, are the likely dominating reasons for higher HMS concentrations when comparing the two areas. Modelled HMS generally had a lesser impact on both the N. Hemispheric and CONUS domains (generally contributing < 1 ug/m3), with a few anomalous exceptions. Modelled HMS in HCMAQ runs primarily appeared in Europe and China, where wintertime HMS has previously been predicted (Moch et al., 2020). On the CONUS domain, HMS concentrations were predicted to by high during major PM events (including but not limited to wildfires) which illuminates the importance of HMS during high atmospheric loadings of both $SO_2$ and HCHO.

Across all modelled domains and episodes, the TMI_sens model run predicted the highest HMS concentrations and may represent an upper bound on modelled HMS concentrations. Ultimately, more resolved measurements of speciated $PM_{2.5}$ that can separate out HMS and $SO_4^{2-}$ (Campbell et al., 2022) can help discern their relative contributions to $PM_{2.5,sulf}$ mass and help constrain future modelled heterogeneous sulfur chemical kinetics.



### 4.3 PM$_{2.5,sulf}$ formation pathways of interest during cold and dark episodes

In addition to the inclusion of both heterogeneous SO$_4^{2-}$ and HMS formation in CMAQ, we determined which PM$_{2.5,sulf}$ formation pathways are the most important given ionic strength, pH, and temperature regimes characteristic of dark and cold

conditions. Across both the Fairbanks and CONUS domains in the Base_Het during the wintertime, the most prevailing heterogeneous SO$_4^{2-}$ formation pathway was the TMI-catalyzed O$_2$ pathway (Fig. 2, 4, S11). In the TMI_sens E1 in Fairbanks, however, this formation pathway was the third most important behind HMS formation and the NO$_2$ pathway (Fig. S3). Although the modelled pH for the TMI_sens ranged between 3-6 for Fairbanks and North Pole and for both episodes (Fig. S4) which included the optimal pH for this pathway (pH=4.2; (Ibusuki and Takeuchi, 1987)), the dampening of this

pathway can mostly be attributed to the extremely cold temperatures (modelled average -30º C or 243º K), which drastically lower the $k_{chem}$.

TMI_sens modelled aerosol pH was seen to be least acidic in comparison to all of the other model simulations, especially in North Pole (Fig. S4). As noted before, HMS was the largest contributor to secondary PM$_{2.5,sulf}$ formation at North Pole, the

formation (and loss) rates of which increase with increasing pH (Ervens et al., 2003; Kok et al., 1986) (Fig. 2). Aerosol pH and ALW calculations in ISORROPIA II only consider inorganic species. Organic species (e.g., organic acids) may also increase aerosol acidity (Zuend et al., 2011; Zuend and Seinfeld, 2012), and therefore the predicted aerosol pH in the TMI_sens might represent an overprediction. Aerosol pH for the Base_Het, TMI_NO2_sens, and All_Ionic model simulations were similar at both North Pole and Fairbanks with both sensitivity simulations predicting slightly higher pH

than the Base_Het simulation during E1 and slightly lower pH during E2.

The impacts of increased ionic strength were explored with respect to the NO$_2$, O$_3$, and H$_2$O$_2$ oxidation pathways (note that ionic strength inhibition of the TMI-O$_2$ pathway is included in the Base_Het simulations as well). Ionic strength impacts added to the NO$_2$ pathway (in the TMI_NO2_sens) had the highest impact on the formation of SO$_4^{2-}$ via this pathway (Fig.

S3 and S7 c and d) in the Fairbanks domain for both episodes. Although the ionic strength for this pathway was bounded at 1.14 M, an increase in aerosol ionic strength from 0.1 M to the upper bound of 1.14 M increased the $k_{chem}$ for this pathway by ~2 orders of magnitude (Chen et al., 2019). Ionic strength impacts on the H$_2$O$_2$, and O$_3$ heterogeneous sulfur oxidation pathways had minimal impact during the wintertime PM episodes in Fairbanks and North Pole, only accounting for ≤~0.2 µg/m$^3$ in the All_Ionic model simulation (Fig. S3 and Fig. S7). These particular pathways were not assumed to be prolific

given the lack of photochemistry during this episode; however with an ionic strength change from 0 to 5 M, the 3$^{rd}$-order aqueous-phase rate coefficient for the H$_2$O$_2$ heterogeneous sulfur pathway can increase by more than 40% regardless of pH or temperature (Maaß et al., 1999; Millero et al., 1989). The ionic strength used to calculate the ionic strength effect factor for this $k_{chem}$ was limited to a maximum of 5 M; however recent studies have observed significant ionic-strength enhancement up to 14 M (Liu et al., 2020). For SO$_4^{2-}$ formation via heterogeneous oxidation by O$_3$, the $k_{chem}$ for this



pathway can increase by ~80% with an ionic strength increase from 0.1 to 0.8 M (Lagrange et al., 1994; Song et al., 2021b) at temperatures characteristic of Fairbanks winters, however these effects were not seen due to the lack of ozone modelled given the dark and cold conditions.

**5 Conclusion**

Air quality modelling of secondary sulfate has traditionally only included in-cloud aqueous- and gas-phase $SO_2$ oxidation

pathways, often resulting in underpredictions of observed $PM_{2.5,sulf}$, especially during the cold and polluted conditions characteristic of wintertime PM and haze events (ADEC, 2017; Gao et al., 2016). In this study we implemented heterogeneous sulfur chemistry in aerosol liquid water in CMAQ to resolve model-measurement gaps in $PM_{2.5,sulf}$ concentrations during extreme wintertime PM episodes in and around Fairbanks, Alaska. We compared modelled $PM_{2.5,sulf}$ ($SO_4^{2-}$+HMS) concentrations to sulfate measurements at several measurement sites (under the assumption that HMS may be

included in the sulfate observations (Dovrou et al., 2019)). Negative model bias improved in Fairbanks during winter and fall with these updates and also improved in Beijing, another location known to experience wintertime haze events. When applied more broadly to larger domains and other seasons, the update also resolved underestimations of $PM_{2.5,sulf}$ concentrations both in the United States and globally, however mostly did not have a huge impact when over domains that were not as dark and cold. HMS was found to be an important contributor to PM2.5,sulf mass during dark and cold episodes,

however, to better understand the ratio of sulfate to HMS, more observations of HMS are necessary. Recently, the ALaskan Pollution and Chemical Analysis (ALPACA) field campaign was conducted in and around Fairbanks during January-February 2022 and will offer observations to elucidate important sulfate and HMS formation pathways in the area and better characterize source apportionment of $PM_{2.5,sulf}$. Ultimately, HMS and sulfate formed via the TMI-catalysed $O_2$ and $NO_2$ pathways proved to be the most important to $PM_{2.5,sulf}$ formation pathways in Fairbanks and North Pole and require further

investigation in the context of $PM_{2.5,sulf}$ control strategies. Finally, while this study aims to include ionic strength, pH, and temperature impacts on $PM_{2.5,sulf}$ formation in CMAQ, the ionic strength, pH and temperature ranges under which Henry's law, reaction rates, and other coefficients and parameters were derived experimentally and may not be representative of concentrated aerosol water or the extreme environment of subarctic and arctic wintertime conditions. Laboratory studies are needed to extend the bounds of these parameters and determine rate expressions appropriate for the concentrated conditions

characteristic of aerosol water.

*Disclaimer:* The views expressed in this article are those of the authors and do not necessarily represent the views or policies of the US Environmental Protection Agency, Alaska Department of Environmental Conservation, or the University of North Carolina at Chapel Hill.




*Author Contributions:* HOTP, KF, SG and NB were responsible for conceptualization and funding acquisition. RG was responsible for meteorology development and prepared meteorology inputs. GP was responsible for emissions development and prepared emissions inputs. SF, KF and HOTP were involved with the methodology. SF and KF were involved with the software. KF lead the model validation. SF, KF and DH were involved in the investigation and SF and KF lead the formal analysis. SF, KF, and DH were responsible for data curation. KF and HOTP were both involved with supervision and project administration. SF, KF, RG and GP were involved in writing the original draft. All co-authors contributed to writing-review and editing.

*Competing Interests:* The authors declare that they have no conflict of interest.

*Acknowledgments:* This work was supported by the U.S. Environmental Protection Agency Office of Research and Development. This research was supported in part by an appointment to the U.S. Environmental Protection Agency (EPA) Research Participation Program administered by the ORISE through an interagency agreement between the U.S. DOE and the U.S. Environmental Protection Agency. ORISE is managed by ORAU under DOE contract number DE-SC0014664. The authors would like to thank Wyat Appel and Kristen Foley for guidance in statistical analysis and assessing model performance statistics, Christian Hogrefe for assistance with EMEP data and HCMAQ, Fahim Sidi for help with running and updating CMAQ, David Wong and Emma D'Ambro for helpful discussions, Robert Kotchenruther for initial project conceptualization, and Kristen Foley and Chris Nolte for Internal Review of this work.

*Code availability:* CMAQ is available at https://github.com/USEPA/CMAQ and CMAQv5.3.2 is archived at https://doi.org/10.5281/zenodo.4081737 (U.S.EPA, Office of Research and Development, 2020). The exact CMAQ code and data used in this work will be available at https://catalog.data.gov/dataset.

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
