# Peer review of "Predicted impacts of heterogeneous chemical pathways on particulate sulfur over Fairbanks, Alaska, the N. Hemisphere, and the Contiguous United States"

_EGUsphere, 2024_

## Author Comment (AC2)

**Addressing Review Comment 1**

Reviewer comments are reproduced in black. Responses are in blue. Updates to the manuscript are shown with underline (addition) or strikeout (removal).

Many models are not able to reproduce high sulfate concentrations, and do not consider heterogeneous chemistry in aerosol droplets. This paper examines sulfate and HMS formation in aerosol droplets as a possible cause for model underestimation. This is interesting work which I recommend for publications upon completion of some minor revisions.

Thank you for the time it took to review our paper, your kind words, and suggestions!

1. Sentence starting on line 41 is hard to read due to length and many parentheses. I suggest splitting it into two or more sentences.

The following change has been made:

40

Sulfate ($SO_4^{2-}$), often a major component of $PM_{2.5}$ in Fairbanks and North Pole (ADEC, 2017) as well as globally (Snider et al., 2016), can be emitted directly (primary) or formed secondarily via atmospheric  oxidation of sulfur dioxide ($SO_2$). Known secondary $SO_4^{2-}$ formation processes include but are not limited to gas phase oxidation of $SO_2$ (Calvert et al., 1978), particle surface oxidation of $SO_2$ (Clements et al., 2013; Wang et al., 2021), and aqueous-phase oxidation of inorganic

45 sulfur species with oxidation number 4 (S(IV) = $SO_2 \cdot H_2O$ + $HSO_3$ + $SO_3$) (secondary) (Hoffmann and Calvert, 1985; Ibusuki and Takeuchi, 1987; Lagrange et al., 1994; Lee and Schwartz, 1983a; Maahs, 1983; Maaß et al., 1999; Martin and Good, 1991; McArdle and Hoffmann, 1983). Aside from contributing directly to $PM_{2.5}$ mass, $SO_4^{2-}$ can facilitate the formation of other $PM_{2.5}$ species as a reactant (Brüggemann et al., 2020; Huang et al., 2019; Huang et al., 2020; Surratt et al., 2010), by increasing aerosol water uptake (Kim et al., 1994; Nguyen et al., 2014), and by altering aerosol acidity (Li et al.,

50 2022; Pye et al., 2020).

**2. Line 100: write out CONUS**

The following change has been made:

100 horizonal resolutions. Two historical wintertime PM episodes were simulated for a finely resolved (1.33 km) domain centered over Fairbanks, Alaska, winter, and summer periods over the contiguous United States (CONUS) (12 km) during 2016, and the 2015-2016 winter season over the N. hemisphere (108 km) to investigate the impacts of these updates for different chemical regimes, domains, and seasons. Changes to $SO_4^{2-}$, HMS, and $SO_4^{2-}$ + HMS ($PM_{2.5,sulf}$) predictions were tracked with each update (i.e., for (1) adding heterogeneous sulfur reactions and (2) adding ionic strength effects), and model

105 performance was evaluated with available observations. This study aims to better understand the impacts that heterogeneous sulfur chemistry parameterizations may have on predicted $PM_{2.5,sulf}$ concentrations and whether the additional chemistry can resolve $SO_4^{2-}$ underpredictions in cold and dark conditions.

3. Methods: It's unclear how ALW and pH were calculated. Please state explicitly where these numbers (for example the pH and ALW in line 331) come from.

The thermodynamic equilibrium model, ISOROPPIA (Fountoukis and Nenes, 2007) was used to calculate aerosol pH and ALW. In response to this suggestion, we've included a small paragraph stating this in section 2.3:

| | | | 1996) |
|---|---|---|---|

ᵃ Aqueous phase concentrations of S(IV) are calculated similarly to the Base_Het, TMI_sens, and TMI_NO2_sens but with
195   ionic-strength dependent equilibrium coefficients.

Inorganic ion concentrations in CMAQ are passed to the thermodynamic equilibrium model, ISORROPIA II, to calculate aerosol pH and ALW then passed back (Fountoukis and Nenes, 2007). In addition to ALW calculated based off inorganic ion water activity, ALW associated with organic aerosols are also estimated in CMAQ via hygroscopicity parameters (Pye et al.,
200   2017).

**2.4 Model base case and sensitivity simulations**

Several CMAQ configurations were used here to understand the impacts of adding heterogeneous sulfur chemistry, ionic strength, and the use of alternative pseudo-first order rate expressions. A base case CMAQ simulation ("Base") was completed using in-cloud $SO_4^{2-}$ formation from aqueous oxidation by $H_2O_2$, $O_3$, PAA, MHP, and via TMI-catalyzed $O_2$ of $SO_2$ and gas-
205   phase oxidation of $SO_2$ by OH (Fahey et al., 2017; Sarwar et al., 2013).

325   Out of all of the secondary $PM_{2.5,sulf}$ formation pathways that are enhanced during dark cold conditions (TMI-catalyzed $O_2$, $NO_2$, and the formation of HMS), the leading secondary $SO_4^{2-}$ formation pathway in the Base_Het is the TMI-catalyzed $O_2$ oxidation pathway in ALW (Fig. 2). The first order condensed phase rate constant ($k_{chem}$) of this pathway is lower than that of the $k_{chem}$ for $NO_2$ by almost 2 orders of magnitude for average modeled conditions characteristic of Fairbanks and North Pole for E1 (pH = 3.83, [Fe(III)] = 0.24 M,  [Mn(II)] = 0.002 M, [$SO_2$] = 20 ppb, [NO2] = 20 ppb, [$SO_4^{2-}$] = 3 μg/m³, [ALW]

4. In figures 1 and 3, the concentrations of the species are hard to see because the text partially covers it. Stating the domain size would also be helpful here.

We shifted the labels a little outside of the area of interest and made the font size smaller so that concentrations can be better seen and included the domain size as well in the caption:

[Figure]

**Figure 1:** Episode average sulfate (a), and HMS (c) concentrations in the Base simulation along with daily max differences in sulfate (b), and HMS (d) concentrations between the Base_Het and Base CMAQ simulations over Fairbanks and North Pole AK for episode 1 (from January 25th, 2008, to February 11th, 2008). HMS formation was not included in Base CMAQ (i.e., HMS = 0 in the Base simulation). Domain size is 264.67 km by 264.67 km with a grid cell resolution of 1.33 km by 1.33 km.

5. In Figure 1a, it seems there's a high (~1 ug/m3) background of sulfate surrounding the Fairbanks and North Pole area, which seems strange. I would expect near-zero sulfate concentrations in these areas because there is very little anthropogenic activity.

Thank you for pointing this out. These concentrations are attributed to background conditions. While the background concentrations are not 0, they are not quite ~1 µg/m$^3$ and this is easier to see with a discrete color bar. We made this change to the plots and the background sulfate concentration for our base run is ~0.6 µg/m$^3$:

(a)

[Figure]

While most boundary conditions in modeling studies are seasonal averages, we used hourly-resolved boundary conditions for 2008 from the EQUATES project (USEPA, 2021). We have included a sentence in section 2.4 detailing this:

280 Gas-phase chemistry was simulated using the CB6r3 mechanism (Luecken et al., 2019) and aerosol dynamics were simulated using the aero7 module. Boundary conditions for the Fairbanks domain were sourced from the EPA's Air QUAlity TimE Series Project (EQUATES) (USEPA, 2021). The sulfur tracking method (STM) (which is documented at https://github.com/USEPA/CMAQ/blob/main/DOCS/Users_Guide/CMAQ_UG_ch12_sulfur_tracking.md and used in

285 (Fahey and Roselle, 2019)) was extended to include the new heterogeneous sulfur chemical pathways in order to track the contributions of each chemical reaction, primary emissions, and initial and boundary conditions to modelled $SO_4^{2-}$ (Appel et al., 2021).

6. Line 358: HSO3 and SO3 should have their charges written out like sulfate ($SO_4^{2-}$). Check for other mentions of HSO3 and SO3 in the paper.

These typo's have been addressed in this line and throughout the paper.

7. In Figure 7, is there any explanation for the major differences on Dec 13 and 27? I think this should be discussed due to the large discrepancy between model and measurements.

When looking into the cause for these differences, we realized that we had accidentally mis-matched model and observed time points by 1 day. We have resolved this and now this is what Fig. 7 should look like:

[Figure]

We have replaced this figure in the paper and the model-measurement gap for Dec. 13[th] is resolved slightly. We have also updated the model performance metrics in the text.

There still remains a large discrepancy between model output and observations for Dec. 22$^{nd}$. Our hemispheric simulations (while our heterogeneous chemistry updates were included) did not include the sulfur tracking method tags for our new pathways and therefore contributions from each pathway were not tracked. The contribution of each pathway can be potentially inferred with looking at precursor oxidant concentrations. In this newly created figure (Fig. S10), the dominant the $PM_{2.5,sulf}$ peak modeled concentrations trend with peak coincidental $SO_2$, $NO_2$, and TMI concentrations:

[Figure]

I have included discussion of the Dec 22$^{nd}$ discrepancy as well:

The Base_Het and all additional sensitivity runs predicted higher $PM_{2.5,sulf}$ at this grid cell than the Base model simulation and reduced modelled mean bias by 2.97 µgS/m³ (model mean bias with Base was -4.25 µgS/m³ and mean bias with Base_Het was -1.38 µgS/m³) (Fig. 7). Despite the overall improvement in model performance in the Base_Het simulation, a substantial gap in modeled and measured $PM_{2.5,sulf}$ still exists on Dec. 22$^{nd}$. Daily averaged modeled $SO_2$, $NO_2$, HCHO and TMI concentrations (from Base HCMAQ, representing a lower-bound for $SO_2$ consumption) for this time period show that peak $PM_{2.5,sulf}$ concentrations coincide with the co-occurrence of heightened $SO_2$+TMI+$NO_2$ concentrations (Fig. S10). On Dec. 22$^{nd}$, while $SO_2$ concentrations reach a daily average of ~22ppb, $NO_2$ and TMI concentrations are ~1/2 the concentrations they are Dec 7$^{th}$-8$^{th}$.

8. Line 716: ALPACA should be Alaska Layered Pollution And Chemical Analysis. You may want to cite this paper as well https://doi.org/10.1021/acsestair.3c00076

Thank you for this suggestion, we have included this citation.

Fountoukis, C., and Nenes, A. (2007). ISORROPIA II: a computationally efficient thermodynamic equilibrium model for $K^+–Ca^{2+}–Mg^{2+}–NH_4^+–Na^+–SO_4^{2-}–NO_3^-–Cl^-–H_2O$ aerosols. *Atmos. Chem. Phys., 7*(17), 4639-4659. doi:10.5194/acp-7-4639-2007

USEPA. (2021). *EQUATESv1.0: Emissions, WRF/MCIP, CMAQv5.3.2 Data -- 2002-2019 US_12km and NHEMI_108km*. Retrieved from: https://doi.org/10.15139/S3/F2KJSK

---

## Author Comment (AC3)

Addressing Review Comment 2

Reviewer comments are reproduced in black. Responses are in blue. Updates to the manuscript are shown with underline (addition) or strikeout (removal).

The discrepancy between field-observed sulfate concentrations during haze episodes and the values simulated by air quality models has garnered significant attention over the past two decades. Many scientists believe the traditional mechanism for S(IV) reaction in cloud chemistry is inadequate. Therefore, the multiphase and heterogeneous chemistry of S(IV) compounds has been a particularly intriguing topic in atmospheric chemistry. However, there is a lack of models that incorporate the dominant mechanisms into air quality models for comparison, and very few simulations specifically focus on the impact of ionic strength on reaction rates. The key methodological contribution of this paper is the implementation of a model developed by the authors using CMAQ to simulate the conversion of $SO_2$ to sulfate and HMS, yielding accurate results in Alaska. I had a few minor reservations in my reading, but I still highly recommend this article for publication in Atmospheric Chemistry and Physics.

Thank you for the time it took to review our paper, your kind words, and suggestions!

Here are my suggestions.

1. Line 18: The definition of "heterogeneous" needs clarification. In my understanding, Heterogeneous processes can be categorized as surface chemistry, while multiphase chemistry generally refers to reactions occurring in the liquid phase. (DOI:10.1126/science.276.5315.1058, DOI: 10.5194/acp-23-9765-2023)

Thank you for attaching the above articles. We use the heterogeneous reactive uptake parameterization outlined in Hanson et al., (1994) to parameterize the multistep process of diffusion of a reactant towards a particle, dissolution in the particle, and reaction in the particle. Based on the rate of reaction vs diffusion of the precursor in the particle, the uptake may scale with surface area (fast reaction relative to particle diffusion) or volume (slow reaction relative to particle diffusion). The abstract has been reworded to indicate we model the process as heterogeneous reactive uptake:

**Abstract.** A portion of Alaska's Fairbanks North Star Borough was designated as nonattainment for the 2006 24-hour PM$_{2.5}$ National Ambient Air Quality Standard (NAAQS) in 2009. PM$_{2.5}$ NAAQS exceedances in Fairbanks mainly occur during the dark and cold winters, when temperature inversions form and trap high emissions at the surface. Sulfate ($SO_4^{2-}$), often the second largest contributor to PM$_{2.5}$ mass during these wintertime PM episodes, is underpredicted by atmospheric chemical transport models (CTMs). Most CTMs account for primary $SO_4^{2-}$, and secondary $SO_4^{2-}$ formed via gas-phase oxidation of sulfur dioxide ($SO_2$) and in-cloud aqueous oxidation of dissolved S(IV). Dissolution and reaction of $SO_2$ in aqueous aerosols<s>,</s> is generally not <s>often</s> included in CTMs<s>,</s> but can be represented as heterogeneous reactive uptake and may help better represent the high $SO_4^{2-}$ concentrations observed during Fairbanks winters. In addition, hydroxymethanesulfonate (HMS), a particulate sulfur species sometimes misidentified as $SO_4^{2-}$, is known to form during Fairbanks winters. Heterogeneous formation of $SO_4^{2-}$ and HMS in aerosol liquid water (ALW) was implemented in the Community Multiscale Air Quality (CMAQ) modeling

And then also included this at the end of the introduction:

In this paper we describe the implementation of heterogenous sulfur chemistry in ALW in the Community Multiscale Air Quality (CMAQv5.3.2) modeling system (USEPA, 2020), leading to additional $SO_4^{2-}$ and HMS formation. We refer to this chemistry as heterogeneous given the use of the heterogeneous framework (Hanson et al., 1994). Heterogeneous sulfur chemistry pathways implemented include the oxidation of dissolved S(IV) species by $H_2O_2$, $O_3$, PAA, MHP, $TMI-O_2$, $NO_2$, and the in-aerosol aqueous formation of HMS. In addition to heterogeneous chemistry updates, ionic strength effects were added to condensed-phase rate expressions and Henry's law coefficients of some species. The updated model was applied for several time periods and for different domains and horizonal resolutions. Two historical wintertime PM episodes were

2. Line 39: Please give the meaning of "2006 24-hour PM2.5 NAAQS".

We have included a footnote to define this:

suggested that secondary $SO_4^{2-}$ may be efficiently produced in aerosol liquid water (ALW) (Cheng et al., 2016, Pan et al., 2020; Liu et al., 2020). Hygroscopic $PM_{2.5}$ (both inorganic and organic) can increase ALW content (Nguyen et al., 2014; Petters and Kreidenweis, 2007; Pye et al., 2017), which can facilitate secondary $SO_4^{2-}$ formation (Zhang et al., 2021a), enhancing $SO_4^{2-}$ concentrations in a positive feedback loop – which is particularly important during high relative humidity haze events (Cheng et al., 2016; Song et al., 2021b; Wang et al., 2016; Wang et al., 2014).

[1] The United States Clean Air Act requires EPA to set National Ambient Air Quality Standards (NAAQS) for six pollutants including fine particulate matter (PM2.5) and to periodically review those standards. In 2006, EPA updated the NAAQS for PM2.5 concentrations averaged over a 24-hour time period. This updated standard requires the calculation of the 98[th] percentile of daily (24-hour) PM2.5 concentrations for three years, and that the average of those three 98[th] percentile concentrations be at or below a threshold of 35 ug/m3. For simplicity, we refer to this as the 2006 24-hour PM2.5 NAAQS.

3. Line 109: The introduction provides detailed information on specific reaction mechanisms in the gas phase and clouds. This paper suggests presenting the new mechanisms introduced here in detail and reconfirming the roles of heterogeneous and multiphase processes. It is recommended that the specific mechanisms introduced in this paper be listed in detail in this section and that the issues related to heterogeneous and multiphase processes be reconfirmed.

We have included the specific heterogeneous pathways in the last paragraph of the introduction

In this paper we describe the implementation of heterogenous sulfur chemistry in ALW in the Community Multiscale Air Quality (CMAQv5.3.2) modeling system (USEPA, 2020), leading to additional $SO_4^{2-}$ and HMS formation. We refer to this

100 chemistry as heterogeneous given the use of the heterogeneous framework (Hanson et al., 1994). Heterogeneous sulfur chemistry pathways implemented include the oxidation of dissolved S(IV) species by $H_2O_2$, $O_3$, PAA, MHP, TMI-$O_2$, $NO_2$, and the in-aerosol aqueous formation of HMS. In addition to heterogeneous chemistry updates, ionic strength effects were added to condensed-phase rate expressions and Henry's law coefficients of some species. The updated model was applied for several time periods and for different domains and horizonal resolutions. Two historical wintertime PM episodes were

105 simulated for a finely resolved (1.33 km) domain centered over Fairbanks, Alaska, winter, and summer periods over the contiguous United States (CONUS) (12 km) during 2016, and the 2015-2016 winter season over the N. hemisphere (108 km) to investigate the impacts of these updates for different chemical regimes, domains, and seasons. Changes to $SO_4^{2-}$, HMS, and

4. Line 130: How should the boundary problem of ionic strength (I) in aerosol water be addressed? Although this is mentioned later, the I values used here are based on maximum boundaries tested in laboratory tests. However, in actual aerosol during haze events, I can often reach several tens of M, which is significantly higher than the few M observed in laboratory conditions. Considering the potential exponential growth of the enhancement factor (EF) with increasing ionic strength (I), the intensity of aerosol ions may significantly impact the reaction rate. Of course, these are merely my thoughts and discussions. The authors do not need to address this issue directly, but they could consider it further in their outlook or future work.

This is a good and an important topic for future modeling work! In representing reactions that can occur in dark, cold, haze conditions, we would be eager to see the ionic strength experimental bounds extended for the $NO_2$ and TMI-catalyzed $O_2$ aqueous oxidation pathways. We looked into incorporating the higher ionic strength bounds published in Liu et al., (2020) for secondary sulfate formation via $H_2O_2$, however, this pathway was not a dominant sulfate formation pathway in the episodes and contexts in Alaska with minimal photochemistry (Liu et al., 2020). Nonetheless, this can be investigated in the future.

5. Lines 320-324: It is recommended that HMS use a different color bar range than sulfate. Using a maximum value of 5, for instance, results in nearly zero HMS concentration, and the spatial distribution of HMS is not effectively captured in Figure 1c. The same issue is observed for the figures 3, 6, 8, and 10.

Yes. The reason why the concentrations appear nearly zero in these plots is because they are zero. HMS is not an included species in Base CMAQ (neither formed in ALW nor in cloud liquid water). To evade confusion, we included this information in the figure caption for all of the aforementioned figures:

[Figure]

325

Figure 1: Episode average sulfate (a), and HMS (c) concentrations in the Base simulation along with daily max differences in sulfate (b), and HMS (d) concentrations between the Base_Het and Base CMAQ simulations over Fairbanks and North Pole AK for episode 1 (from January 25th, 2008, to February 11th, 2008). HMS formation was not included in Base CMAQ (i.e., HMS = 0 in the Base simulation). Domain size is 264.67 km by 264.67 km with a grid cell

330 resolution of 1.33 km by 1.33 km.

Nonetheless, to pair better with the counter difference plot, we have constrained the color bars and made all discrete instead of continuous for all of these figures and Fig 12.

6. Lines 331-332: What does atmospheric acidity, particularly aerosol pH, look like in this context? It is suggested that the authors consider incorporating pH into the exploration of dominant pathways to help explain why TMI is dominant in Alaska.

To clarify, the pH referred to in this line is the episode-averaged modelled aerosol pH. The overall heterogeneous production rate for the TMI-O₂ pathway in the Base_Het is pH dependent in that the effective Henry's law coefficient for SO2 is pH dependent, however, its $k_{chem}$ is actually pH independent (Table 1) (Martin and Good, 1991). We have made the following modifications:

Out of all of the secondary PM$_{25\_sulf}$ formation pathways that are enhanced during dark cold conditions (TMI-catalyzed O$_2$, NO$_2$, and the formation of HMS), the leading secondary SO$_4^{2-}$ formation pathway in the Base_Het is the TMI-catalyzed O$_2$

335    oxidation pathway in ALW (Fig. 2). The first order condensed phase rate constant ($k_{chem}$) of this pathway is lower than that of the $k_{chem}$ for NO$_2$ by almost 2 orders of magnitude for average modeled conditions characteristic of Fairbanks and North Pole for E1 (aerosol pH = 3.83, [Fe(III)] = 0.24 M, [Mn(II)] = 0.002 M, [SO$_2$] = 20 ppb, [NO2] = 20 ppb, [SO$_4^{2-}$] = 3 µg/m$^3$, [ALW] = 6 µg/m$^3$, and Temp = 243K) (Fig. S2) and is ~ 1 order of magnitude higher than that for HMS formation in ALW. Despite the NO$_2$ $k_{chem}$ being higher, however, the TMI-catalyzed O$_2$ heterogeneous rate of sulfate formations rate limiting

340    stepis dependent upon is SO$_2$ partitioning into the particle, as Fe and Mn are both aerosol species, and simulated dark conditions reduce the conversion of Fe$^{3+}$ to Fe$^{2+}$ from daytime photochemical reactions (Alexander et al., 2009; Rao and Collett, 1998; Shao et al., 2019). The effective Henry's law coefficient for SO$_2$ increases with pH, while the Henry's law coefficient for NO$_2$ remains low across the pH spectrum. This and a higher mass accommodation coefficient (by ~2 orders of magnitude) for SO$_2$ compared to NO$_2$ Another potential reasoncontribute to thethe TMI-catalyzed O$_2$ pathway _outcompetinges the NO$_2$ pathways

345    for this model configuration. is due to its mass accommodation coefficient ($\alpha$, Eq. 2) being higher than that for the NO$_2$ pathway by ~2 orders of magnitude. The TMI-catalyzed O$_2$ heterogeneous reactive uptake pathway also outcompetes the H$_2$O$_2$ and O$_3$ heterogeneous reactive uptake pathways due to low photochemical activity with the dark conditions of this domain and episode.

We incorporate a pH and temperature dependent (Ibusuki and Takeuchi, 1987) in our sensitivity runs (TMI_sens, TMI_NO2_sens, and ALL_IONIC) to explore the effects of acidity and temperature on the $k_{chem}$ of this pathway and the entire sulfate and HMS formation system. We find that this formation pathway no longer dominates, however, it is difficult to say whether this is aerosol pH or temperature driven.

Using a back-of-the-envelope excel calculation, when decreasing aerosol pH from 4 to 3, the $k_{chem}$ for the TMI pathway decreases by ~81%, however, when decreasing the temperature from 243K to 233K (a decrease in temperature that is within range for Fairbanks winters), the $k_{chem}$ for the TMI pathway decreases by ~77%. Therefore, this particular formation pathway is sensitive to both temperature and pH. We made this change to better clarify this takeaway in the discussion:

**4.3 PM$_{2.5,sulf}$ formation pathways of interest during cold and dark episodes**

In addition to the inclusion of both heterogeneous SO$_4^{2-}$ and HMS formation in CMAQ, we determined which PM$_{2.5,sulf}$
680    formation pathways are the most important given ionic strength, pH, and temperature regimes characteristic of dark and cold
conditions. Across both the Fairbanks and CONUS domains in the Base_Het during the wintertime, the most prevailing

heterogeneous SO$_4^{2-}$ formation pathway was the TMI-catalyzed O$_2$ pathway (Fig. 2, 4, S11). In the TMI_sens E1 in Fairbanks,
however, this formation pathway was the third most important behind HMS formation and the NO$_2$ pathway (Fig. S3).
Although the modelled pH for the TMI_sens ranged between 3-6 for Fairbanks and North Pole and for both episodes (Fig. S4)
685    which included the optimal pH for this pathway (pH=4.2; (Ibusuki and Takeuchi, 1987)), the dampening of this pathway can
mostly be attributed to the extremely cold temperatures (modelled average -30° C or 243° K)~~, which drastically lower the
k$_{chem}$~~.

It is also noted, however, that aerosol pH may be overestimated in the TMI_sens given the methods
used to calculate it only consider inorganic aerosol species and PM$_{2.5,sulf}$ concentrations are largely
HMS:

TMI_sens modelled aerosol pH was seen to be least acidic in comparison to all of the other model simulations, especially in
690    North Pole (Fig. S4). As noted before, HMS was the largest contributor to secondary PM$_{2.5,sulf}$ formation at North Pole, the
formation (and loss) rates of which increase with increasing pH (Ervens et al., 2003; Kok et al., 1986) (Fig. 2). Aerosol pH
and ALW calculations in ISORROPIA II only consider inorganic species. Organic species (e.g., organic acids) may also
increase aerosol acidity (Zuend et al., 2011; Zuend and Seinfeld, 2012), and therefore the predicted aerosol pH in the TMI_sens
might represent an overprediction. Aerosol pH for the Base_Het, TMI_NO2_sens, and All_Ionic model simulations were
695    similar at both North Pole and Fairbanks with both sensitivity simulations predicting slightly higher pH than the Base_Het
simulation during E1 and slightly lower pH during E2.

7. Lines 306 and 381: The title 'Time' is not recommended. If you want to highlight the similarities
between sections 3.1.1 and 3.1.2, consider combining the discussions. If the goal is to emphasize the
differences, please choose a title that reflects the unique feature of each section.

The goal is to emphasize the differences and therefore we changed the sub-headings to reflect this:

**3 Results**

310 **3.1 Modelled particulate sulfur enhancement during dark and cold PM episodes in Fairbanks and North Pole, AK**

**3.1.1 pisode 1 (E1 **

The Base simulation average E1 sulfate concentrations around Fairbanks and North Pole, AK are ~ 2 - 3.5 mg/m$^3$ (Fig. 1a and c). Compared to the Base, the Base_Het simulation leads to increased $PM_{2.5,sulf}$ predictions concentrated around the cities of Fairbanks and North Pole as well as the region south of the Tanana River (Figure 1b, d). The additional heterogeneous

315 chemistry in the Base_Het simulation contributes up to an additional 11 µg/m$^3$ of maximum daily $PM_{2.5,sulf}$ compared to the

**3.1.2 pisode 2 (E2 **

390 Sulfate and HMS are known to form efficiently in cloud and fog droplets (Altwicker and Nass, 1983; Boyce and Hoffmann, 1984; Calvert et al., 1978; Clifton et al., 1988; Ibusuki and Takeuchi, 1987; Lee and Schwartz, 1983a; Martin and Good, 1991; McArdle and Hoffmann, 1983). In E1, there was minimal cloud or fog liquid water simulated; however, during E2 (November 4 – November 11, 2008), there were some periods where cloud/fog chemistry impacts on $PM_{25,sulf}$ formation were evident.

395 Compared to E1, $PM_{2.5,sulf}$ concentration enhancements were lower overall during E2. Differences between Base_Het and Base simulations, however, are appreciable during this episode with $PM_{2.5,sulf}$ increasing up to 4.6 µg/m$^3$ across the entire domain (daily maximum difference) (Fig. 3). Enhancements in $PM_{2.5,sulf}$ are mainly driven by increased $SO_4^{2-}$ formation in and around Fairbanks and North Pole; however, simulated HMS concentrations reached up to 4.4 µg/m$^3$ south of the Tanana River (daily

8. Line 649: I was very excited to see the HMS simulation. I'm eager to know whether the modeling of HMS and the multiphase chemistry of sulfate (including the effects of ionic strength) will be included in a future official version of CMAQ.

We plan to incorporate the updates from this work in CMAQv6.0 which as a 2026 target date for release. 😊

Ibusuki, T., and Takeuchi, K. (1987). Sulfur dioxide oxidation by oxygen catalyzed by mixtures of manganese(II) and iron(III) in aqueous solutions at environmental reaction conditions. *Atmospheric Environment (1967), 21*(7), 1555-1560. doi:https://doi.org/10.1016/0004-6981(87)90317-9

Liu, T., Clegg, S. L., and Abbatt, J. P. D. (2020). Fast oxidation of sulfur dioxide by hydrogen peroxide in deliquesced aerosol particles. *Proceedings of the National Academy of Sciences, 117*(3), 1354. doi:10.1073/pnas.1916401117

Martin, L. R., and Good, T. W. (1991). Catalyzed oxidation of sulfur dioxide in solution: The iron-manganese synergism. *Atmospheric Environment. Part A. General Topics, 25*(10), 2395-2399. doi:https://doi.org/10.1016/0960-1686(91)90113-L

Wang, Y., Zhang, Q., Jiang, J., Zhou, W., Wang, B., He, K., Duan, F., Zhang, Q., Philip, S., and Xie, Y. (2014). Enhanced sulfate formation during China's severe winter haze episode in January 2013 missing

from current models. *Journal of Geophysical Research: Atmospheres, 119*(17), 10,425-410,440. doi:https://doi.org/10.1002/2013JD021426

Zheng, B., Zhang, Q., Zhang, Y., He, K. B., Wang, K., Zheng, G. J., Duan, F. K., Ma, Y. L., and Kimoto, T. (2015). Heterogeneous chemistry: a mechanism missing in current models to explain secondary inorganic aerosol formation during the January 2013 haze episode in North China. *Atmos. Chem. Phys., 15*(4), 2031-2049. doi:10.5194/acp-15-2031-2015